# Quantifying black carbon light absorption enhancement by a novel statistical approach

**Cheng Wu[1,2], Dui Wu[1,2,3], Jian Zhen Yu[4,5,6]**

[1]Institute of Mass Spectrometer and Atmospheric Environment, Jinan University, Guangzhou 510632, China

[2]Guangdong Provincial Engineering Research Center for on-line source apportionment system of air pollution, Guangzhou 510632, China

[3]Institute of Tropical and Marine Meteorology, China Meteorological Administration, Guangzhou 510080, China

[4]Division of Environment, Hong Kong University of Science and Technology, Clear Water Bay, Hong Kong, China

[5]Atmospheric Research Centre, Fok Ying Tung Graduate School, Hong Kong University of Science and Technology, Nansha, China

[6]Department of Chemistry, Hong Kong University of Science and Technology, Clear Water Bay, Hong Kong, China

*Corresponding to*: Cheng Wu (wucheng.vip@foxmail.com) and Jian Zhen Yu (jian.yu@ust.hk)

## Abstract

Black carbon (BC) particles in the atmosphere can absorb more light when coated by non-absorbing or weakly absorbing materials during atmospheric aging, due to the lensing effect. In this study, the light absorption enhancement factor, $E_{abs}$, was quantified using one year's measurement of mass absorption efficiency (MAE) in the Pearl River Delta region (PRD). A new approach for calculating primary MAE ($MAE_p$), the key for $E_{abs}$ estimation, is demonstrated using the Minimum R Squared (MRS) method, exploring the inherent source independency between BC and its coating materials. A unique feature of $E_{abs}$ estimation by the MRS approach is its insensitivity to systematic biases in EC and $\sigma_{abs}$ measurements. The annual average $E_{abs550}$ is found to be 1.50±0.48 (±1 S.D.) in the PRD region, exhibiting a clear seasonal pattern with higher values in summer and lower in the winter. Elevated $E_{abs}$ in the rainy summer season is likely associated with aged air masses dominating from marine origin, along with long-range transport of biomass burning influenced air masses from Southeast Asia. Core-shell Mie simulations along with measured $E_{abs}$ and Angstrom absorption exponent (AAE) constraints suggest that in the PRD, the coating materials are unlikely to be dominated by brown carbon and the coating thickness is higher in the rainy season than the dry season.

# 1 Introduction

Originating from incomplete combustion, black carbon (BC) is a crucial constituent of atmospheric aerosols, and is an air pollutant itself, having an adverse health impacts on humans (Suglia et al., 2008). BC has also been recognized as the third most important climate forcer due to its broad light absorbing capability across the UV-Vis-IR spectrum (IPCC, 2013). BC can alter the climate in a variety of ways, including by direct forcing (Bond et al., 2011), affecting cloud cover (Koch and Del Genio, 2010) and precipitation (Tao et al., 2012), reducing the albedo of snow and ice (Hansen and Nazarenko, 2004) and causing surface dimming (Wild, 2011). The climate effects of BC can be global or regional (Ramanathan and Carmichael, 2008). A recent study found BC can modify planetary boundary layer meteorology, and thus enhance local pollution indirectly (Ding et al., 2016). However, due to its variable optical characteristics induced during atmospheric aging, large uncertainties still exist in estimating the radiative forcing from BC. Optical properties of BC can be predicted by knowing the mass concentration, mixing state and size distribution, which collectively serve as the cornerstone for modeling the climate effect of BC. In 3D modeling studies, to conserve computational resources, the mass absorption efficiency (MAE) or mass absorption cross-section (MAC) is widely used to convert black carbon mass concentration to light absorption coefficient ($\sigma_{abs}$). MAE is a quantity to describe the light absorption ability per unit EC mass:

$$\text{MAE } (m^2 \ g^{-1}) = \frac{absorption \ coefficient \ \sigma_{abs} \ (Mm^{-1})}{EC \ mass \ concentration \ (\mu g \ m^{-3})} \qquad (1)$$

As a fundamental input parameter, MAE has a critical impact on BC's radiative forcing estimation in climate modeling studies. Mixing state is one of the governing factors affecting MAE. Light absorption of soot particles is enhanced when coated with non-absorbing materials (Fuller et al., 1999) or weakly absorbing materials (Lack and Cappa, 2010) during atmospheric aging. The coating materials can focus more light onto the soot core through the lensing effect, resulting in elevated MAE (Wang et al., 2017). Strong correlations between MAE and the number/volume fraction of coated particles have been reported in urban areas like Tokyo (Naoe et al., 2009), Shenzhen (Lan et al., 2013) and Xi'an (Wang et al., 2014), implying that the elevated MAE observed at these locations was mainly due to the elevated fraction of coated of soot particles. Total absorption ($\sigma_{abs,t}$) of coated particles can

be separated into two parts: primary absorption ($\sigma_{abs,p}$) due to the uncoated soot core alone, and extra
absorption ($\sigma_{abs,c}$) due to lensing effect of the coating (Bond et al., 2006; Jacobson, 2006; Liu et al.,
2016a) and the presence of secondarily formed brown carbon (BrC) (Lack and Cappa, 2010; Liu et al.,
2016b).

$$\sigma_{abs,t} = \sigma_{abs,p} + \sigma_{abs,c} \qquad (2)$$

The absorption enhancement factor ($E_{abs}$) then can be defined as ratio of the total absorption and
primary absorption coefficients or the corresponding MAE values:

$$E_{abs} = \frac{\sigma_{abs,t}}{\sigma_{abs,p}} = \frac{MAE_t}{MAE_p} \qquad (3)$$

Where $MAE_p$ represents the ratio of $\sigma_{abs,p}/EC$ for uncoated soot particles, similar to the concept of
the primary OC/EC ratio in the EC tracer method:

$$MAE_p = \frac{\sigma_{abs,p}}{EC} \qquad (4)$$

And the MAE of coated BC can be defined as:

$$MAE_t = \frac{\sigma_{abs,t}}{EC} \qquad (5)$$

Thus, elevated MAE induced by coating during atmospheric aging results in an $E_{abs}$ larger than 1.
Previous model studies suggest that absorption by aged soot particles can be 1.5 times greater than
fresh soot (Fuller et al., 1999; Bond et al., 2006). Laboratory studies have demonstrated that soot
particles coated with SOA (Saathoff et al., 2003; Schnaiter et al., 2005; Tasoglou et al., 2017) and
sulfuric acid (Zhang et al., 2008; Khalizov et al., 2009) can increase $E_{abs}$. An artificial coating
experiment by Shiraiwa et al. (2010) found an $E_{abs}$ of 2 for graphite particles growing in diameter from
185 to 370 nm. A laboratory study by McMeeking et al. (2014) found that in the presence of BrC, light
absorption enhancement is more pronounced at the shorter wavelength. A recent chamber study
coupling actual ambient air with seed BC particles implies that the timescale for $E_{abs}$ reaching 2.4 is
only 5 hours in Beijing but 18 hours in Houston (Peng et al., 2016). Field studies conducted in recent
years have also substantiated enhanced light absorption in Canada (Knox et al., 2009; Chan et al.,
2011), US (Lack et al., 2012b), UK (Liu et al., 2015) and Japan (Nakayama et al., 2014; Ueda et al.,
2016). In contrast, field studies in California, US (Cappa et al., 2012) found a weaker light absorption
enhancement (6% on average). A recent study suggests the mass ratio of non-BC content to BC
particles determines the occurrence of the absorption enhancement of black-carbon particles (Liu et
al., 2017).

89         Two approaches are widely used to determine $E_{abs}$ from ambient measurements. The first approach

removes the coating materials on particles physically using a thermal denuder (TD) (Lack et al., 2012a)
or by aerosol filter filtration-dissolution (AFD) (Cui et al., 2016b). The TD approach is briefly
discussed here. Coating materials can be removed by TD at a working temperature around 200 to
300 °C (depending on the charring characteristics of aerosols at the sampling site) to measure $\sigma_{abs,p}$,
which are cycled with measurements of $\sigma_{abs,t}$ (without passing through TD), allowing $E_{abs}$ to be
obtained from the ratio of $\sigma_{abs,t}/\sigma_{abs,p}$ following Eq.3. The major advantage of the TD approach is
its ability to provide highly time resolved measurements (minutes). A photo-acoustic spectrometer
(PAS) is commonly used with TD for detection to satisfy its high time resolution demands. As an in-
situ technique, PAS eliminates the artifacts associated with filter-based methods (Weingartner et al.,
2003; Coen et al., 2010) and is often considered as the reference instrument for light absorption
coefficient determination (Arnott et al., 2003; Arnott et al., 2005). One limitation of the TD approach is
that a universal optimal operation temperature does not exist, leading to a chemical composition
dependent efficiency. If the temperature is too low, the coating cannot be fully removed, and charring
can occur if the TD temperature is too high, leading to biased results. For example, a SP-AMS study
in Toronto found that the efficiency of BC coating removal by TD decreased substantially for wildfire
influenced samples (Healy et al., 2015). Another issue is particle loss due to TD, which can be ~ 20%
and needs to be taken into account (Ueda et al., 2016). It's also worth noting that $MAE_p$ by the TD
approach is different from the $MAE_p$ at the emission source. First, the morphology of thermally
denuded BC particles (compact aggregates) is different from that of freshly emitted BC particles
(chain-like aggregates). Second, most of the coatings is removed with the TD denuded soot particles,
but freshly emitted soot particles usually come with a thin coating of OC formed from condensation
of OC vapors as the temperature drops from engine to the ambient air. As a result, the $MAE_p$ by TD

approach is expected to be lower than the $MAE_p$ of emission source. In this sense, the TD approach may not be a prefect "time machine" to reverse the aging process for $E_{abs}$ determination.

The second approach is the MAE ratio method, which is also stated in Eq. 3. The key to this method is determining an appropriate $MAE_p$ that can represent the MAE from primary emissions. One approach is to adopt the reference $MAE_p$ from the literature but it may fail to represent the actual $MAE_p$ at a specific sampling site, since $MAE_p$ varies temporally and spatially. For example, $MAE_p$ of diesel soot was found to be 7.1 $m^2g^{-1}$ at 532 nm (Adler et al., 2010). A much higher $MAE_p$ (16 $m^2g^{-1}$ at 530 nm) was observed from natural gas flaring (Weyant et al., 2016). $MAE_p$ of biomass burning (BB) samples is highly varied due to a wide range of fuel types and combustion conditions (Reid et al., 2005; Roden et al., 2006). A range from 6.1 to 80.8 $m^2g^{-1}$ was reported for BB $MAE_p$ at 550 nm (Pandey et al., 2016). Without the knowledge of source contributions, it is not feasible to derive a representative $MAE_p$ for $E_{abs}$ estimation. The other commonly used approach is to determine $MAE_p$ from the dependency of MAE on the number fraction of coated soot particles measured by SP2 (Lan et al., 2013). Since MAE (y axis) is positively correlated with the number fraction of coated soot particles (x axis), $MAE_p$ can be determined by extending the regression line to x=0. It is worth noting that this approach provides only a rough approximation of $E_{abs}$ since the parameter used here (coated soot particles number fraction) ignores other main drivers of light absorption enhancement (e.g. coating thickness). As a result, this approach is only valid for a period of measurements, for which coating thickness is relatively constant and the MAE variations are dominated by coated soot particles number fraction.

However, the high cost of the TD-PAS system and SP2 limit the field measurement of $E_{abs}$ around the world. In addition, long-term $E_{abs}$ measurements by a TD-PAS system and SP2 are not easily achieved and rarely reported. On the other hand, an Aethalometer and RT-ECOC analyzer can be effectively deployed for long term measurements and $E_{abs}$ estimation, at a relatively lower cost. In this study, based on one year of hourly MAE measurements (with the field carbon analyzer and Aethalometer) at a suburban site in the Pearl River Delta (PRD) region of China, quantification of $MAE_p$ is demonstrated by a novel statistical approach, the Minimum R squared method (MRS) (Wu and Yu, 2016). The aim of this study is to demonstrate the capability of $E_{abs}$ estimation using a year-

long dataset from cost-effective instrumentation. The seasonal variability of MAE, AAE and $E_{abs}$ in
the PRD region are characterized and their dependency on air mass origin and biomass burning are
discussed. Abbreviations used in this study are summarized in Table 1 for a quick lookup.

## 143    2 Ambient measurements

Sampling was conducted from Feb 2012 to Jan 2013 at the suburban Nancun (NC) site (23° 0'11.82"N,
113°21'18.04"E). NC, situated on the top of the highest peak (141 m ASL) in Guangzhou's Panyu
district, is located at the geographic center of the PRD region, making it a representative location for
average atmospheric mixing characteristics of city clusters in the PRD region. Light absorption
measurements were performed by a 7-λ Aethalometer (AE-31, Magee Scientific Company, Berkeley,
CA, USA). The Aethalometer was equipped with a 2.5 μm cyclone with a sampling flow rate of 4 L
min$^{-1}$. Weingartner's algorithm (Weingartner et al., 2003) was adopted to correct the sampling artifacts
(aerosol loading, filter matrix and scattering effect) rooted in filter based method. A customized
Aethalometer data processing program (Wu, 2017a) with graphical user interface was developed to
perform data correction and detailed descriptions can be found in the SI (The program is available
from https://sites.google.com/site/wuchengust). Details of the Aethalometer setup and data correction
can be found in our previous paper (Wu et al., 2013).
EC mass concentrations were determined by a real time ECOC analyzer (Model RT-4, Sunset
Laboratory Inc., Tigard, Oregon, USA). The sunset carbon analyzer was sampling on hourly cycles at
a flow rate of 8 Lmin$^{-1}$ with a PM$_{2.5}$ sharp-cut cyclone inlet. For each measurement hour, the first
45min were for sample collection and the remaining 15 min for thermal-optical analysis. OC is
volatized first by step-wise temperature ramping in an oxygen-free atmosphere while in the second
stage EC is combusted in the presence of oxygen. Laser transmittance is applied to correct the charring
artifact during the OC stage.
Considering a measurement precision of 5% for the Aethalometer (Hansen, 2005) and 24% for the
RT-ECOC analyzer (Bauer et al., 2009), the propagated relative precision of $E_{abs}$ ($E_{abs,Unc}$) is 35%
following Eq. S1&S2 in the SI. It should be noted that $E_{abs,Unc}$ is mainly attributed to the
measurement precision of EC by the RT-ECOC analyzer. Since the measurement precision of the RT-
ECOC analyzer estimated by Bauer et al. (2009) is obtained from field measurement at an environment
(EC below 1 µg m$^{-3}$) where EC is much lower than the present study (annual average EC 2.66±2.27
µg m$^{-3}$), the $E_{abs,Unc}$ of 35% should be considered as an upper limit for the present study.
Light scattering was measured by an integrating nephlometer (Aurora-1000, Ecotech, Melbourne,
Australia). Water soluble ions were measured by MARGA (The instrument for Measuring AeRosols
and GAses)(ten Brink et al., 2007). Both instruments are equipped with a PM$_{2.5}$ inlet to remove the
coarse particles.

## 2.1 Uncertainties of MAE determination

Two major uncertainties associated with the $\sigma_{abs}$ and EC determination techniques should be taken
into account when comparing MAE across different studies. For the $\sigma_{abs}$ determination technique,
photo-acoustic spectroscopy (PAS) is an in-situ technique free from filter based artifacts, but its
application is limited by its high cost. The filter based optical transmittance method (e.g., Aethalometer
and Multi Angle Absorption Photometer, MAAP) is the most widely used technique around the world,
but data correction is needed to minimize the bias from artifacts due to the loading effect, matrix effect
and scattering effect (Weingartner et al., 2003; Arnott et al., 2005; Schmid et al., 2006; Virkkula et al.,
2007; Coen et al., 2010; Drinovec et al., 2017; Saturno et al., 2017). Besides these artifacts, RH is also
a source of $\sigma_{abs}$ measurement uncertainty. Elevated RH is not only a driving force of increased $\sigma_{abs}$
due to the hygroscopic growth of particles, but also a factor affecting ambient $\sigma_{abs}$ measurements.
Previous studies found $\sigma_{abs}$ by PAS exhibit a systematic decrease when RH>70% (Arnott et al., 2003;
Kozlov et al., 2011). Water evaporation was found as the major cause for the biased PAS $\sigma_{abs}$
measurements under high RH (Raspet et al., 2003; Lewis et al., 2009b; Langridge et al., 2013). Filter-
based measurements are also affected under high RH conditions. For example, Arnott et al. (2003)
observed erratic responses by particle soot absorption photometer (PSAP) as RH changed. The main
reason is traced to the hydrophilic cellulose membrane, which serves to reinforce the quartz filter used
in PSAP. The fibers can swell and shrink as RH changes, causing unwanted light attenuation signal.
The PTFE-coated glass-fiber tape has become available since 2012 for the recent model of
Aethalometer to minimize the RH interference (Magee-Scientific, 2017). A study by Schmid et al.
(2006) reported dependency of PSAP $\sigma_{abs}$ on RH, but found negligible effect of RH on Aethalometer
performance. It is also worth noting that RH in the Aethalometer optical chamber may be lower than
the ambient RH due to the slightly elevated temperature inside the instrument. The magnitude of RH
difference was found similar between different instruments: 20% for the Aethalometer (Schmid et al.,
2006) and 15% for the nephelometer (Guyon et al., 2004). The RH in the Aethalometer optical chamber
was not measured in this study. We expected its level to be slightly lower than the ambient RH. Cappa
et al. (2008) found $\sigma_{abs}$ measurements by PSAP and PAS maintained a high linearity even under high
RH conditions (65-91%). Inter-comparison studies demonstrated that with proper corrections,
Aethalometer $\sigma_{abs}$ measurements agree well with those by PAS (Ajtai et al., 2011). During the inter-
comparison study of an Aethalometer (AE-16) and a PAS in Guangzhou (Wu et al., 2009), good
correlation was found ($R^2$=0.96) as shown in Figure S1. These comparison results imply that the
Aethalometer results are linearly correlated with PAS measurements and RH has a limited interference
on Aethalometer measurements. In our study, careful corrective measures (Wu et al., 2013) are
conducted for the Aethalometer $\sigma_{abs}$ data treatment to minimize these artifacts. But such artifacts
still cannot be fully eliminated.
For the EC determination, different thermal optical analysis (TOA) protocols can impact the
measurement variability and thus MAE. As shown in Table S1, MAE for the same samples at Fresno
varied from 6.1 to 9.3 $m^2\,g^{-1}$, depending on which EC analysis protocol was applied (Chow et al.,
2009). Studies in the PRD found that discrepancies in measured EC by different analysis protocols
could be as large as a factor of 5 (Wu et al., 2012; Wu et al., 2016a), which adds to the uncertainty for
the MAE estimation. In addition, EC by TOA is also different from refractory BC (rBC) reported by
the laser induced incandescence (LII) technique (e.g. single particle soot photometer, SP2). For
example, two studies in Toronto (Knox et al., 2009; Chan et al., 2011) both used the PAS for $\sigma_{abs}$
measurement but different techniques for EC mass determination, resulting in very different MAE
results. LII instruments are usually calibrated with a commercially available surrogate (e.g. fullerene)
since direct calibration with ambient soot is not easy to achieve. Laborde et al. (2012) indicates that
the incandescence response of SP2 exhibits a dependency on soot type (15% between fullerene and
denuded diesel soot particles; 14% between biomass burning and denuded diesel soot particles). Due
to the absence of widely accepted reference materials for EC, the uncertainties in EC determination
will exist in the foreseeable future. All these uncertainties, including the uncertainty of rBC mass
determination by SP2, uncertainty of EC in TOA, the discrepancy between SP2 rBC and TOA EC and
the discrepancy of $\sigma_{abs}$ between filter transmission and photo-acoustic methods, can contribute to the
differences in MAE listed in Table S1.
Systematic bias in MAE (e.g. overestimation of $\sigma_{abs}$ and variability of EC mass by different
TOA protocols) discussed above have little effect on $E_{abs}$ estimation by MRS. As shown in Eq. 3, $E_{abs}$
is the ratio of $MAE_t$ to $MAE_p$ or $\sigma_{abs,t}$ to $\sigma_{abs,p}$, thus most of the bias in EC mass or $\sigma_{abs}$ is
cancelled out during the $E_{abs}$ calculation. More details are discussed in section 4.1.

## 231 3 Methodology

### 232 3.1 MAE$_p$ estimation by MRS from the ambient data

In this section, a new approach for MAE$_p$ estimation is introduced for $E_{abs}$ determination, which
requires the knowledge of differentiating $\sigma_{abs,p}$ and $\sigma_{abs,c}$ portions in $\sigma_{abs,t}$ . The idea of
decoupling $\sigma_{abs,t}$ into $\sigma_{abs,p}$ and $\sigma_{abs,c}$ is conceptually similar to decoupling OC into primary OC
(POC) and secondary OC (SOC) in the EC tracer method as shown in Table 2. In the EC tracer method,
if (OC/EC)$_p$ is known, POC can be determined from OC (Turpin and Huntzicker, 1991). The role of
MAE$_p$ here is similar to the role of (OC/EC)$_p$, the primary OC/EC ratio in the EC tracer method (a
comparison is given in Table 2). If MAE$_p$ (average MAE from primary emission sources) is known,
E$_{abs}$ can be obtained from the ratio of MAE$_t$/MAE$_p$ (Eq. 3). Therefore, the key for E$_{abs}$ estimation is to
derive an appropriate MAE$_p$. It is worth noting that MAE$_p$ here does not represent MAE from a single
or specific primary emission source, instead it reflects an average and effective MAE that has taken
consideration of various primary emission sources. Thus, the MAE$_p$ is conceptually analogous to
(OC/EC)$_p$ in the EC tracer method, in which the primary ratio reflects an overall ratio from primary
emission sources rather than from a single primary source.
The Minimum R squared method (MRS) explores the inherent independency between
pollutants from primary emissions (e.g., EC) and products associated with secondary formation
processes (e.g., SOC, $\sigma_{abs,c}$) to derive the primary ratios (e.g., (OC/EC)$_p$, MAE$_p$) in the EC tracer
method (Wu and Yu, 2016). When applying MRS for light absorption enhancement estimation, MRS
is used to explore the inherent independency between EC and $\sigma_{abs,c}$, which is gained during
atmospheric aging after emission. An example of $MAE_p$ estimation by MRS is shown in Figure 1.
Firstly, the assumed $MAE_p$ value is varied continuously in a reasonable range (0.01 to 50 $m^2$ $g^{-1}$ as
shown in Figure 1). Then at each hypothetical $MAE_p$, $\sigma_{abs,c}$ can be calculated by Eq. 6 (a combination
of Eq. 2&4) using EC and $\sigma_{abs,t}$ from ambient measurements.
$$\sigma_{abs,c} = \sigma_{abs,t} - MAE_p \times EC \tag{6}$$

256        Accordingly, for each hypothetical $MAE_p$, a correlation coefficient value ($R^2$) of $\sigma_{abs,c}$ vs.

EC (i.e., $R^2(\sigma_{abs,c}$, EC)) can be obtained. The series of $R^2(\sigma_{abs,c}$, EC) values (y axis) are then plotted
against the assumed $MAE_p$ values (x axis) as shown by the red curve in Figure 1. The physical meaning
of this plot can be interpreted as follows. The $\sigma_{abs,p}$ is the fraction of light absorption owing to
primary emitted soot particles. As a result, $\sigma_{abs,p}$ is well correlated with EC mass. In contrast, the
$\sigma_{abs,c}$ is the fraction of light absorption gained by the lensing effect of the coating on particles after
emission. The variability of $\sigma_{abs,c}$ mainly depends on the coating thickness of the soot particles.
Consequently, $\sigma_{abs,c}$ is independent of EC mass. Since variations of EC and $\sigma_{abs,c}$ are independent,
the assumed $MAE_p$ corresponding to the minimum $R^2(EC, \sigma_{abs,c})$ would then represent the most
statistically probable $MAE_p$ of the tested dataset.

266        A computer program (Wu, 2017b) in Igor Pro (WaveMetrics, Inc. Lake Oswego, OR, USA)

was developed to facilitate MRS calculation with a user friendly graphical user interface. Another two
Igor Pro based computer programs Histbox (Wu, 2017c) and Scatter Plot (Wu, 2017d) are used for
generating histograms, box plots and scatter plots (with Deming regressions) presented in this study.
Detailed descriptions of these computer programs can be found in the SI and the computer programs
are available from https://sites.google.com/site/wuchengust.
**3.2 Mie simulation**

273        It can be informative to model a single soot particle using Mie theory (Bohren and Huffman,

1983) and understand the theoretical range and variability of the soot particle's optical properties.
Three types of mixing state are widely employed for parameterization: internal mixing, external
mixing and core-shell. To better represent the real situation (coating due to the aging process), a core-
shell model is considered in the Mie calculation (Figure S2), which is more realistic than a volume
mixture model (Bond et al., 2006). An aerosol optical closure study in the North China Plain (NCP)
found that the core-shell model can provide better performance than assuming purely internal mixing
and external mixing (Ma et al., 2012). A morphology study using Scanning Transmission X-ray
Microscopy found that core-shell is the dominating mixing state in ambient samples (Moffet et al.,
2016). It should be noted that the core-shell model assumption still has its own limitations. A single
particle soot photometer (SP2) study by Sedlacek et al. (2012) reported a negative lag time between
the scattering and incandescence signals in samples influenced by biomass burning, implying   a near
surface location of soot relative to non-absorbing materials. Near surface type mixing of soot has also
been observed in Tokyo, but accounted for only 10% of total mixed soot containing particles (Moteki
et al., 2014). Considering the domination of core-shell type particles in the ambient environment, the
core-shell assumption in our optical model is sufficient to approximate the real situation.
As shown in Figure S2, fresh emitted soot particles are chain-like aggregates of small spheres
(30~50 nm). After the aging process, soot particles are coated with organic and inorganic materials.
Sufficient evidence has shown that the coating not only results in particle size growth, but also makes
the soot core become more compact due to its collapse (Alexander et al., 2008; Zhang et al., 2008;
Lewis et al., 2009a), especially under high RH conditions (Leung et al., 2017). A recent study by Pei
et al. (2017) shown that filling of void space within the agglomerate is the first step of the
morphological transformation of soot particles in atmospheric aging, leading to a spherical soot core.
Since the spherical like core and shell favor Mie simulation, both core and shell are considered as
spheres in the Mie calculation.
To investigate the spectrum properties of soot particles, 11 wavelengths (370, 405, 470, 520,
532, 550, 590, 660, 781, 880 and 950 nm) are considered in calculations to cover wavelengths in the
most frequently used absorption measurement instruments. A refractive index (RI) of $1.85 - 0.71i$ is
adopted for soot core (Bond and Bergstrom, 2006) and 1.55 for non-absorbing coating (clear shell) in
the Mie calculation for all wavelengths. Studies suggest a group of organic matter (OM), known as
Brown Carbon (BrC), can absorb solar radiation at UV wavelengths (Kirchstetter et al., 2004). Thus,
a BrC coating (brown shell) scenario is also considered in Mie simulation following the wavelength-

dependent RI suggested by Lack and Cappa (2010), which ranges from 1.55-0.059i (370 nm) to 1.55-0.0005i (950 nm). A modeling study by Bond et al. (2006) indicates that absorption amplification is not sensitive to the RI, thus the result below is not expected to be sensitive to the RI variability. Due to the spherical assumption of the BC core, a constant particle density is adopted for simplicity instead of size dependent particle density. But it is worth noting that in reality, the effective density of soot varies with particle size due to the morphology change during particle aging (Tavakoli and Olfert, 2014; Dastanpour et al., 2017). Both core diameters ($D_{core}$) and shell diameters ($D_{shell}$) are constrained in the range of 10 ~ 3000 nm in the model simulations. The Mie calculations are implemented with a customized program (Wu, 2017e) written in Igro Pro (WaveMetrics, Inc. Lake Oswego, OR, USA) and it is available from https://sites.google.com/site/wuchengust. It should be noted that the core-shell type mixing state of particles is still rare in 3D atmospheric models like WRF-Chem (Matsui et al., 2013; Nordmann et al., 2014) due to computational cost limitation.

### 3.2.1 Mie modeled absorption angstrom exponent (AAE)

Absorption Angstrom Exponent (AAE) is a widely used parameter that describes the wavelength dependence of aerosol light absorption (Moosmuller et al., 2011), which can be written explicitly as

$$AAE(\lambda_1, \lambda_2) = -\frac{\ln(\sigma_{abs,\lambda1}) - \ln(\sigma_{abs,\lambda2})}{\ln(\lambda_1) - \ln(\lambda_2)} \qquad (7)$$

It is well known that ambient soot particles exhibit an AAE close to unity (Bond, 2001). Modeled variability in $AAE_{470-660}$ of bare soot particles is shown in Figure S3. For soot particles with $D_{core}$ <200 nm, $AAE_{470-660}$ is very close to 1 and decreases significantly for particles with $D_{core}$ >200 nm. Considering a typical $D_{core}$ of fresh emitted soot particles smaller than 200 nm (Rose et al., 2006; China et al., 2013), the model results confirm the frequently observed AAE close to 1 from ambient measurements (Kirchstetter et al., 2004). Modeled variability in $AAE_{470-660}$ of soot particles coated by non-absorbing substances (clear shell) and weakly absorbing materials (brown shell) is shown in Figure 2. Elevated AAE to ~2 is observed in the clear shell scenario (Figure 2a and 3b) for the most probable soot core particle sizes (<200 nm), which agrees well with a previous model study (Lack and

Cappa, 2010), implying that elevated AAE cannot be exclusively attributed to mixing with BrC. AAE
elevation is more pronounced in the brown shell scenario. For soot particles with $D_{core}$ <200 nm, brown
shell $AAE_{470-660}$ can easily reach 3 for a coating of $D_{shell}/D_{core}$=3 (Figure 2c and 2d). These high AAE
results are consistent with the previous model study (Lack and Cappa, 2010) and could partially
explain the high AAE observed in measurement studies (Kirchstetter et al., 2004; Hoffer et al., 2006),
since the presence of externally mixed BrC particles also contribute to the wavelength dependent light
absorption.
**3.2.2 Mie modeled single scattering albedo (SSA)**
Variability in modeled $SSA_{525}$ of soot particles coated by non-absorbing substances and weakly
absorbing materials (e.g. BrC) is shown in Figure S4. For particles with $D_{core}$<200 nm and $D_{shell}/D_{core}$
<3, the SSA increases gradually (up to ~0.9) with a thicker coating and behaves similarly between
clear shell and brown shell scenarios.
**3.2.3 Mie modeled mass absorption efficiency (MAE)**
MAE is a useful indicator for soot mixing state. Variability in MAE of bare soot particles as a
function of particle size at a wavelength of 550 nm is illustrated in Figure S5. The magnitude of MAE
is sensitive to the soot density assumption, especially for particles <200 nm (Figure S5), but the overall
trend of particle size dependency is similar between different density scenarios. MAE peaks at a
particle size of 200 nm and decreases dramatically for larger particles. In our MAE calculation, a soot
density of 1.9 g $cm^{-3}$ is adopted, as suggested by Bond and Bergstrom (2006). The purpose of adopting
constant density is to simplify the MAE calculation. It should be noted that the effective density of
soot core is highly variable in ambient environments. For example, a study in Beijing (Zhang et al.,
2016b) found a value of 1.2 g $cm^{-3}$. A recent chamber study found the effective density of soot can
evolve from 0.43 to 1.45 g $cm^{-3}$ during aging as coated by m-Xylene oxidation products (Guo et al.,
2016). A study by a single-particle aerosol mass spectrometer in Guangzhou found the effective
density of soot increased with particle size in the range of 400 to 1600 nm (Zhang et al., 2016a). The
MAE of coated particles from different core/shell diameter combinations are shown in Figure S6. For
thickly coated particles, the MAE in the clear shell scenario varied as $D_{shell}/D_{core}$ increased, but the
MAE of brown shell scenario increased quasi-monotonously with $D_{shell}/D_{core}$.

### 3.2.4 Mie modeled light absorption enhancement factor ($E_{abs}$)

$E_{abs}$ is a better indicator for soot mixing state than MAE since it does not rely on the soot density
assumption and is more suitable for comparing Mie simulations with ambient measurements. Modeled
variability in $E_{abs}$ of soot particles coated by non-absorbing substances and weakly absorbing materials
(e.g. BrC) is shown in Figure 3a and 3c respectively. $E_{abs}$ is not only sensitive to the core/shell diameter
combination, but also behaves very differently on the clear and brown shell assumptions. For the clear
shell scenario, when $D_{coat}/D_{core}$ <2, $E_{abs}$ does not exceed 2 for particles with different soot core sizes,
but for the same $D_{coat}/D_{core}$, a larger soot core size yields a higher $E_{abs}$ (Figure 3b, cross-sections of
Figure 3a). If $D_{coat}/D_{core}$ >2, $E_{abs}$ could be 3 to 5 for particles with a soot core smaller than 200 nm, but
for particles with a soot core larger than 200 nm, the $E_{abs}$ is limited to ~2 as shown in Figure 3b. For
the brown shell scenario, $E_{abs}$ increased quasi-monotonically with $D_{coat}/D_{core}$, and this trend is similar
for different soot core sizes (Figure 3d). The main reason behind is that in the brown shell scenario,
both lensing effect and BrC absorption contribute to $E_{abs}$. As shown in Figure S7, the BrC absorption
contribution to total $E_{abs}$ strongly depends on coating thickness and is insensitive to soot core diameters.
When the coating is relatively thin (<5 nm for $\lambda$@370 nm, <15 nm for $\lambda$@550 nm and <40 nm for
$\lambda$@880 nm), BrC absorption contribution to the total $E_{abs}$ is less than 20%. As the coating increases to
a certain level (~15 nm for $\lambda$@370 nm, ~35 nm for $\lambda$@550 nm and ~90 nm for $\lambda$@880 nm), BrC
absorption contribution is comparable to the lensing effect contribution, each contributing ~50% to
the total $E_{abs}$. When the BrC coating is sufficiently thick (>30 nm for $\lambda$@370 nm, >90 nm for $\lambda$@550
nm and >110 nm for $\lambda$@880 nm), BrC absorption dominates the $E_{abs}$ contribution. As a result, if BrC
coating is indeed present in ambient samples, a strong wavelength dependent $E_{abs}$ could be observed,
since a BrC coating of 30 nm would be enough to induce a large amount of detectable $E_{abs}$ in the UV
range. Another major difference between the clear and brown shell scenarios is that, for thickly coated
particles (e.g. $D_{coat}/D_{core}$ >2), the brown shell can yield a much higher $E_{abs}$ than the clear shell.

383       Both primary soot size distribution and coating thickness can affect the absorption

enhancement of ambient BC particles. Ambient measurements by LII found soot particle number and
mass modes peaking at 110 nm and 220 nm, respectively, in the PRD (Huang et al., 2011). A study in
Shanghai found similar results (70 nm for number concentrations and 200 nm for mass
concentrations)(Gong et al., 2016). Considering that the LII technique is specific for BC mass
determination which is independent of BC mixing state, the size distribution reported by LII can
represent the size distribution of the BC core. A study using a Micro Orifice Uniform Deposit Impactor
(MOUDI) found a EC mass size distribution in the PRD exhibiting three modes peaking at ~300, ~900
and ~5000 nm (Yu et al., 2010), implying a substantial coating of BC particles, and a diameter
amplification of 3. BC sizing by LII is based on volume equivalent diameter (VED), while MOUDI is
based on aerodynamic diameter. As a result, these two techniques do not necessarily yield similar sizes,
even for the bare soot particles. The conversion between these two types of diameters involves the
knowledge of particle density and morphology (drag force). A recent closure study on BC mixing state
in the PRD region suggests $\sigma_{abs}$ is dominated by coated soot particles in the range of 300~400 nm
(Tan et al., 2016). Considering the dominant BC core distribution measured by SP2 (110 nm), the
upper limit of $E_{abs}$ in the PRD is roughly estimated as ~2 for the clear shell scenario (Figure 3b).
**4 Results and discussions**
**4.1 Annual measurement statistics**
The frequency distribution (log-normal) of $\sigma_{abs550}$ is shown in Figure 4a, with an annual average (±1
S.D.) of 42.65±30.78 $Mm^{-1}$. A log-normal distribution is also found in the EC mass concentration
(Figure 4b), with an annual average of 2.66±2.27 $\mu g\ m^{-3}$. Figure 4c demonstrates the yearlong
frequency distribution of $MAE_{550}$ at the NC site. The annual average $MAE_{550}$ is 18.75±6.16 $m^2\ g^{-1}$ and
the peak (±1 S.D.) of the lognormal fit is 15.70±0.22 $m^2\ g^{-1}$. A good correlation is observed between
$\sigma_{abs}$ and EC mass ($R^2$=0.92) as shown in Figure 4d, and the color coding indicates a MAE dependency
on RH, which agrees with a study in Xi'an(Wu et al., 2016b). Annual average $AAE_{470-660}$ is 1.09±0.13
(Figure S8a), indicating that soot is the dominant absorbing substance in the PRD and the brown shell
scenario shown in the Mie simulation is unlikely to be important. Annual mean $SSA_{525}$ is 0.86±0.05
(Figure S8c), similar to previous studies in the PRD (Jung et al., 2009; Wu et al., 2009). For
comparison purpose, MAE measured at original wavelength and MAE scaled to 550 nm following the
$\lambda^{-1}$ assumption are both shown in Table S1. The MAE comparisons discussed below are MAE at 550
nm. $MAE_{550}$ by previous studies at various locations was found to cover a wide range, from 5.9 to 61.6
$m^2\,g^{-1}$. Annual average observed $MAE_{550}$ at NC (18.75 $m^2\,g^{-1}$) is higher than many studies shown in
Figure 5, e.g., Shenzhen (Lan et al., 2013), Beijing (Yang et al., 2009), Mexico city (Doran et al., 2007)
and Fresno (Chow et al., 2009).
As shown in Figure 1, the annual average $MAE_{p,550}$ estimated by MRS is 13 $m^2\,g^{-1}$. $MAE_p$ by
MRS represents the $MAE_p$ at the emission source, which is different from the $MAE_p$ by the TD
approach for two reasons. First, the morphology of thermally denuded BC particles (compact
aggregates) is different from that of freshly emitted BC particles (chain-like aggregates). Second, most
of the coatings are removed for TD denuded soot particles, but freshly emitted soot particles usually
come with a thin coating of OC formed from condensation of OC vapors as the temperature drops from
the flame to the ambient air. As a result, the MRS-derived $MAE_p$ is expected to be higher than the
$MAE_p$ by the TD approach. The estimated $MAE_{p,550}$ is higher than a previous study in Guangzhou
(7.44 $m^2\,g^{-1}$) (Andreae et al., 2008), but comparable to Xi'an (11.34 $m^2\,g^{-1}$) (Wang et al., 2014) and
Toronto (9.53~12.57 $m^2\,g^{-1}$) (Knox et al., 2009). The annual average $E_{abs550}$ by MRS following Eq. 3
is estimated to be 1.50±0.48 (mean ± 1 S.D.).
As mentioned in section 1, the definition of $MAE_p$ by the TD approach is different from the
$MAE_p$ of emission source. The TD $MAE_p$ is expected to be slightly lower than the $MAE_p$ of emission
source. Therefore, the corresponding $E_{abs}$ are slightly different and it should be cautioned when
comparing MRS-derived $E_{abs}$ with $E_{abs}$ by the TD approach and Mie simulations. The $E_{abs}$ could vary
by location, depending on the coating thickness and size distribution of the primary aerosols. After
undergoing atmospheric aging, the $E_{abs}$ can be increased during transport from emission source to rural
areas. The magnitude of the $E_{abs}$ found at the NC site is comparable to other locations such as Boulder
(Lack et al., 2012a) (1.38), London (Liu et al., 2015) (1.4), Shenzhen (Lan et al., 2013) (1.3), Yuncheng
(Cui et al., 2016b) (2.25), Jinan (Chen et al., 2017) (2.07) and Nanjing (Cui et al., 2016a) (1.6) and is
higher than studies in California (Cappa et al., 2012) (1.06), as listed in Table 3. Spectrum $E_{abs}$ are
calculated from 370 to 950 nm as shown in Figure S9. $E_{abs}$ in the PRD exhibits a weak wavelength
dependence, with slightly higher $E_{abs}$ at the shorter wavelength (e.g. $E_{abs370} = 1.55 \pm 0.48$) and is
relatively lower in the IR range (e.g. $E_{abs950} = 1.49 \pm 0.49$).

## 441 4.2 Monthly characteristics of MAE, AAE and SSA

Monthly variations of $MAE_{550}$ at the NC site are shown in Figure 6a and Table S2, revealing distinct
patterns of higher $MAE_{550}$ in summer and lower in winter. On the other hand, $AAE_{470-660}$ is lower in
summer and higher in winter (Figure 6b and Table S3). Monthly $SSA_{525}$ varied from 0.83 to 0.90
without a clear seasonal pattern, as shown in Figure S10 and Table S4. $MAE_{p,550}$ estimation for
individual months is shown in Figure 6a (the purple line) and monthly $E_{abs550}$ is calculated accordingly
following Eq. 3 (Figure 6c). $E_{abs550}$ shows clear seasonal variations, with higher values from April to
August (1.52~1.97 as shown in Table S5) and relatively lower values from September to March
(1.24~1.49). The highest enhancement is found in August (1.97). Factors affecting variation of $E_{abs550}$
are discussed in the following sections, including air mass origin and biomass burning.

## 451 4.3 The effect of air mass origin

It's of interest to understand the seasonal variations of optical properties in the PRD. Hourly backward
trajectories for the past 72 hours were calculated using NOAA's HYSPLIT (Hybrid Single Particle
Lagrangian Integrated Trajectory, version 4) model (Draxier and Hess, 1998) from Feb 2012 to Jan
2013 as shown in Figure S11. Cluster analysis was conducted using MeteoInfo (Wang, 2014). By
examining the total spatial variance (TSV), the number of clusters was determined to be four as shown
in Figure S12. Cluster 1 (C1) represents continental air masses from the north, accounting for 44.4%
of total trajectories. C2 (22.8%) represents marine air masses coming from the South China Sea. C3
represents air masses from the east (Taiwan island). C4 (15.8%) represents transitional air masses
coming from the east coastline of China. As shown in Figure 7, $E_{abs550}$ from C2 (1.78) is higher than
other clusters (1.30 – 1.42). Further Wilcoxon-Mann-Whitney tests show that $E_{abs550}$ from C2 is
significantly higher than $E_{abs550}$ from C1, C3 and C4 (Figure S13), implying that particles from the
South China Sea cluster is likely more aged than other clusters. Air mass origin in the PRD is
dominated by C2 from Apr to Aug (Figure S14a) as a result of the South China Sea monsoon in the
rainy season. In contrast, the dry season is ruled by continental air masses from the north (C1) due to
the influence of the northeast monsoon. $E_{abs550}$ from C2 varied from 1.67 to 2.19, but was always
higher than $E_{abs550}$ from C1 and C3 during the rainy season (Figure S14b). As a result, the domination
of aged air mass from the vast ocean is one of the reasons for the much higher $E_{abs550}$ found in the rainy
season.

## 470   4.4 The effect of biomass burning

Biomass burning (BB) and vehicular emission are the two major sources of soot particles. BC
from biomass burning emission, depending on the fuel type and burning condition, may have a higher
OC/EC ratio and a thicker coating, resulting in a higher MAE than vehicular emission (Shen et al.,
2013; Cheng et al., 2016) . In this study, the influence of BB on optical properties is investigated using
the $K^+$/EC ratio as a BB indicator. As shown in Figure 8, $MAE_{550}$ is positively correlated with the
$K^+$/EC ratio, which exhibits a clear seasonal pattern that is higher in the rainy season and lower in the
dry season (Figure S15a). Southeast Asia has the highest fire emission density globally due to the high
biofuel consumption along with frequent fire activity in this region (Aouizerats et al., 2015), making
Southeast Asia a large contributor to BC emissions (Jason Blake, 2014). During the rainy season when
oceanic wind prevails, BC from BB emission in Southeast Asia can reach PRD through long range
transport (LRT), resulting in an elevated $K^+$/EC ratio and $MAE_{550}$. The Deming regression intercept
(11.89) in Figure 8 represents the MAE without the BB effect. This non-BB $MAE_{550}$ (11.89 $m^2\,g^{-1}$) is
only slightly lower than $MAE_{p,550}$ (13 $m^2\,g^{-1}$) obtained in section 4.3, implying that a large fraction of
$MAE_{p,550}$ could not be explained by the BB source. Additional evidence was obtained through
examining regression relationships of $MAE_{p,550}$ with $K^+$/EC month-by-month (Figure S15b).
Correlation of monthly $MAE_{p,550}$ vs. $K^+$/EC ratio yield a $R^2$ of 0.23 (Figure S15c). In contrast, a much
higher correlation ($R^2$=0.58) was observed (Figure S15d) between $MAE_{p,550}$ and non-BB $MAE_{550}$ (i.e.,
$K^+$/EC intercepts from Figure S15b). These results imply that BB is one of the contributors to the
$MAE_{p,550}$ variations, but unlikely the dominating one.
Many studies have found that BB influenced samples exhibit elevated AAE due to the presence
of wavelength dependent light absorbing substances like BrC and HUmic-LIke Substances (HULIS)
(Kirchstetter et al., 2004; Hoffer et al., 2006; Sandradewi et al., 2008; Herich et al., 2011; Pokhrel et
al., 2017). It is of interest to investigate whether elevated AAE observed in the PRD during the dry
season is associated with BB influence. As shown in Figure S16, $AAE_{370-470}$ and $AAE_{470-660}$ did not
correlate with the BB indicator, $K^+/EC$ ratio. These results suggest that the elevated AAE observed in
the PRD wintertime is unlikely to be dominated by the BB effect. Beside the independency between
$AAE_{470-660}$ and $K^+/EC$ ratio, the measured $AAE_{470-660}$ range also implies that BB is not the major
driving force of $AAE_{470-660}$ variations. The limited light absorption contribution from BrC in RPD
region is observed in a recent study (Yuan et al., 2016) , which suggest an upper limit of BrC
contribution of 10% at 405 nm in the winter time using the AAE approach. As discussed in our Mie
simulation (section 3.1) and a previous study (Lack and Cappa, 2010), coating of non-absorbing
materials onto soot particles can increase AAE up to 2. Since the monthly average $AAE_{470-660}$ in
wintertime did not exceed 1.2 (Table S3), the variations of $AAE_{470-660}$ in the PRD are more likely
associated with coatings rather than the contribution of BrC. The results also imply that attempts on
BrC absorption attribution for the PRD dataset presented in this study could be risky, considering that
elevation of AAE is actually dominated by coating (Lack and Langridge, 2013).

## 4.5 Implications for mixing state

Quantitative direct measurements of BC mixing state and coating thickness are still challenging.
SP2 can estimate the coating thickness using a lag-time approach or a Mie calculation approach can
be employed, but both methods have a limited range in coating thickness and uncertainties arise from
the assumptions made during the retrieval. For example, recent studies found that the mass equivalent
diameter of soot core measured by SP2 could be underestimated due to density assumptions (Zhang et
al., 2016b). Although size distribution measurement is not available in this study, clues of mixing state
still can be derived from bulk measurements of optical properties. As discussed in section 4.4.1,
elevated $E_{abs550}$ observed in the rainy season is associated with aged air masses from a marine origin.
To probe the possible mixing state difference between dry and rainy season, $E_{abs550}$, $SSA_{525}$ and
AAE$_{470-660}$ are used to narrow down the possible core-shell size range as shown in Figure S17. Monthly
averages with one standard deviation of AAE$_{470-660}$, SSA$_{525}$ and E$_{abs550}$ are used as constraints to extract
the intersecting core-shell size range from Figure 2a, Figure S4 and Figure 3a. January and August
data are used to represent two different scenarios: elevated AAE$_{470-660}$ (1.19±0.11) with lower E$_{abs550}$
(1.31±0.32) in dry season and low AAE$_{470-660}$ (1.04±0.09) with elevated E$_{abs550}$ (1.97±0.71) in rainy
season. The results show that January and August have a very different core-shell size range: in
January, the core and shell range are 100 ~ 160 nm and 120 ~ 250 nm, respectively; in August, the
core and shell range are 120 ~ 165 nm and 170 ~ 430 nm, respectively. This confirms again that the
soot particles in the rainy season are likely to have a thicker coating than in the dry season.
**5 Caveats of the MRS method in its applications to ambient data**
**5.1 Impact of measurement biases**

528        It should be noted that the E$_{abs}$ estimation approach is insensitive to the systematic MAE bias

(e.g. systematic overestimation of $\sigma_{abs}$ and variability of EC mass by different TOA protocols)
discussed in section 2.1, because systematic bias in EC mass or $\sigma_{abs}$ is cancelled out in the E$_{abs}$
calculation (Eq. 3), since E$_{abs}$ is the ratio of $\sigma_{abs,t}$ to $\sigma_{abs,p}$. To investigate the performance of the
MRS approach in response to systematic bias in EC and $\sigma_{abs}$, two simple tests are conducted as shown
in Figures S18 and S19 by adding systematic biases to the original data. The one-year measurement
data of $\sigma_{abs550}$ and EC are used as original data. Test A represents a situation when $\sigma_{abs}$ is
overestimated and EC is underestimated. The biased data are marked as $\sigma'_{abs550}$ and EC' respectively,
as shown below:
$$\sigma'_{abs550} = \sigma_{abs550} \times 2 \tag{8}$$
$$EC' = EC \times 0.7 \tag{9}$$
As a result, the average MAE$_{550}$ changed from 18.75 to 53.58 m$^2$ g$^{-1}$ and MAE$_p$ changed from 13 to 37
m$^2$ g$^{-1}$ (Figure S18). However, E$_{abs}$ by ratio of averages remain the same (1.44).
In Test B, EC by different TOA protocols are compared to investigate the effect of different EC
determination approaches while $\sigma_{abs550}$ remains unchanged. EC by IMPROVE TOR protocol is
calculated from NIOSH TOT EC following an empirical formula for suburban sites derived from a 3-
year OCEC dataset in PRD (Wu et al., 2016a):

$$EC_{IMP\_TOR} = 2.63 \times EC_{NSH\_TOT} + 0.05 \tag{10}$$


As shown in Figure S19, $MAE_{550}$ changed from 18.75 to 7.02 $m^2 g^{-1}$ and $MAE_p$ changed from 13 to 5
$m^2 g^{-1}$, but $E_{abs}$ remain almost the same (1.40). Result of Test B implies that although EC is
operationally defined, the discrepancy of EC between TOA protocols did not weaken the role of EC
serving as a tracer for primary emissions in MRS application. These examples demonstrate that
systematic biases in $\sigma_{abs550}$ and EC have no effects on $E_{abs}$ estimation by the MRS approach.
Study by Cheng et al. (2016) found two distinct types of biomass smoke behave differently on the
biases of filter based $\sigma_{abs}$ measurement. The bias in the first type can be explained by a nearly
constant correction factor, which is similar to the situation discussed in Test A. The bias in the second
type shows an apparent OC/EC dependence. Test C is carried out to investigate this situation, i.e.,
examining the impact of sample-dependent bias as a function of $E_{abs}$. Unlike the proportional bias in
Test A and B that is the same for all data points, the bias in Test C depends on the $E_{abs550}$ of individual
samples, which are parametrized by Eqs. (11) and (12).

$$\sigma'_{abs550} = \sigma_{abs550} + \sigma_{abs550} \times (k \times E_{abs550} - k) \tag{11}$$


$$EC' = EC - EC \times (k \times E_{abs550} - k) \tag{12}$$


As shown in Eqs. (11) and (12), the positive bias of $\sigma_{abs550}$ and negative bias of EC are proportional
to $E_{abs550}$. The magnitude of $E_{abs550}$-dependent bias is regulated by the factor k. Since $\sigma'_{abs550}$ and EC'
are biased in different directions, resulting a further amplification in MAE biases, which could be
considered as the extreme case. As shown in Figure S20, for k=10% (corresponding to a bias of 10%
when $E_{abs}$=2), the bias of MRS-derived $E_{abs}$ is very small (1%). For k=20%, the MRS-derived $E_{abs}$
changes from 1.44 to 1.66, leading to a bias of 15%. These results imply that if the measurement bias
follows the same form as demonstrated in Test C, the bias is not negligible but still acceptable. If the
impact only affects $\sigma_{abs}$ or EC rather than impacting both, the bias is expected to be smaller than the
estimation shown in Test C.
It should be noted that the parameterization scheme shown in Eqs. (11) and (12) is only for
demonstration purpose from a conceptual perspective and it does not necessarily represent the real-
world measurements. There is a lack of quantitative understanding of this impact. For example, Lee et
al. (2007) used artificially fabricated EC samples with OC coatings to evaluate the impact of coating
on OC/EC analysis. Biases were observed, nevertheless, the results were linearly correlated with the
true OC and EC values with a high $R^2$ (>0.9), implying that the biases in that specific study were
dominated by systematic biases rather than the coating-dependent bias. Further studies are needed to
better characterize and parameterize this impact if filter-based techniques are used for $\sigma_{abs}$ and EC
determination in the MRS approach.

## 5.2 Impact of semi-volatile organic carbon

Light absorption contribution due to semi-volatile organic carbon (SVOC) from wood combustion was
reported to be negligible in the visible range and around 10-20% at 360 nm (Chen and Bond, 2010).
On the other hand, OCEC analysis can be affected by SVOC (Subramanian et al., 2004), either by
positive artifacts through adsorption of SVOC onto quartz filters, or by negative artifacts through
evaporation of SVOC due to the gas-particle re-equilibrium down stream of VOC denuder. Positive
artifacts can be minimized by the installation of a VOC denuder which is widely adopted in RT-OCEC
measurements (Bae et al., 2004; Bauer et al., 2009). A typical negative artifact of 10% is expected and
can be corrected by backup filters (Subramanian et al., 2004). There was evidence to show that SVOC
could affect OC/EC split in thermal/optical analysis (Cheng et al., 2009). However, the bias in EC
caused by the OC/EC split drift due to SVOC is systematic, making it falls into the scenario discussed
in Test B. As a result, the impact from SVOC on $E_{abs}$ estimation by MRS is expected to be small.

## 5.3 Impact of mineral dust

The presence of mineral dust (MD) could affect both $\sigma_{abs}$ and EC determination. If MD is externally
mixed with soot particles, the light absorption from MD could be miscounted as $\sigma_{abs}$ enhancement,
leading to the overestimation of of $E_{abs}$. If the light absorption signal from MD is sufficiently strong
(e.g. AAE>2), $\sigma_{abs}$ by MD and BC can be separated by the AAE approach suggested by Fialho et al.
(2005). Additionally, the presence of substantial MD in samples has several impacts on the EC
determination by thermal analysis. First, if the samples are not pre-treated with acid, the carbonated
carbon could be misidentified as EC, resulting over-estimation of EC (Chow et al., 1993). The acid
treatment is only available for off-line OC/EC analysis and not yet practical for the RT-OCEC analyzer.
Second, metal oxides in MD can lead to premature EC oxidation in the helium stage of OC/EC analysis,
leading to underestimation of EC (Wang et al., 2010; Bladt et al., 2012). The lack of a parameterization
scheme for correcting the EC loss due to MD makes it improper to use the biased EC as a primary
tracer. For these reasons, $E_{abs}$ estimation by MRS is not recommended for samples strongly influenced
by MD.

## 5.4 Impact of BrC

The data in this study is dominated by BC absorption that did not show much influence from
BrC. However, extra care should be taken if the samples exhibit substantial BrC signature (e.g.
AAE>2). Such situations are equivalent to the two-source scenarios discussed in our previous paper
on the MRS method (Wu and Yu, 2016) and the major findings are described below. Two types of
two-source scenarios are considered: two correlated primary sources (scenario A) and two independent
primary sources (scenario B). In scenario A in which both BC and primary BrC are dominated by BB,
using BC as a solo tracer to calculate the primary ratio ($MAE_p$) still works. In scenario B in which BC
and primary BrC are independent, using BC alone to determine a single primary $MAE_p$ could lead to
a considerable bias in $E_{abs}$ estimation. Alternatively, if a reliable primary BrC tracer is available, the
corresponding $MAE_{p,BrC}$ can be determined by MRS. With the knowledge of $MAE_{p,BrC}$ and $MAE_{p,BC}$,
light absorption by BC and BrC can be calculated separately and the $E_{abs}$ can be determined using Eq.

616   (13) :

$$E_{abs} = \frac{\sigma_{abs,t}}{\sigma_{abs,p,BC} + \sigma_{abs,p,BrC}} = \frac{\sigma_{abs,t}}{MAE_{p,BC} \times EC + MAE_{p,BrC} \times BrC} \tag{13}$$

However, the implementation of Eq.13 is challenging due to the complexity in the chemical
composition of BrC. For example, a recent study found that the 20 most absorbing BrC chromophores
account for ~50% BrC light absorption and there is not a single compound contributing more than 10%
(Lin et al., 2016), making it difficult to choose a single compound as the BrC tracer. In addition, time
resolved measurement of BrC chromophores has yet to emerge. As a result, for scenario B (sample
AAE>2 & primary BrC variations independent of BC), estimation of $E_{abs}$ by MRS is not practical at
this stage due to the lack of required input data. Using BC alone to determine a single primary $MAE_p$
could lead to a considerable bias and should be avoided.

## 6 Conclusions

In this study, a novel statistical approach is proposed and its application on ambient data is
demonstrated using one-year hourly OC and EC data coupled with Aethalometer measurements.
Unlike conventional $E_{abs}$ determination approaches that require expensive instrumentation (e.g. TD-
PAS, VTDMA, SP2), this new approach employs widely deployed instruments (field carbon analyzer
and Aethalometer). The key of this new approach involves calculating $MAE_p$ by the Minimum R
Squared (MRS) method (Wu and Yu, 2016). The MRS method opens up a new approach to investigate
the long-term trend of $E_{abs}$ that was rarely studied by the TD approach. It is found that $E_{abs}$ estimation
by MRS is insensitive to systematic biases in EC and $\sigma_{abs}$ measurements. The annual average
$MAE_{p,550}$ estimated by MRS is 13 $m^2\,g^{-1}$ and annual average $MAE_{550}$ is $18.75\pm6.16\ m^2\,g^{-1}$, suggesting
an annual average enhancement factor ($E_{abs550}$) of $1.50\pm0.48$ in the PRD region. This value is within
the upper limit of $E_{abs}$ (~2) by core-shell Mie simulations considering the typical soot size distribution
and coating thickness in the PRD.
Both $MAE_{p,550}$ and $E_{abs}$ show distinct seasonal variations, implying the complexity of soot
particle mixing state variations in this region. The elevated summertime $E_{abs550}$ in the PRD is found to
be associated with the domination of aged air masses from the South China Sea, along with the long-
range transport of biomass burning influenced air masses from Southeast Asia. Core-shell size ranges
narrowed down by $E_{abs550}$ and $AAE_{470-660}$ constraints suggest that soot particles in the rainy season are
likely to have thicker coatings than in the dry season.

## Data availability

OC, EC, inorganic ions and $\sigma_{abs}$ data used in this study are available from corresponding authors upon request.

## Acknowledgements

This work is supported by the National Natural Science Foundation of China (41605002, 41475004). We gratefully acknowledge the Fok Ying Tung Foundation for funding to the Atmospheric Research Center at HKUST Fok Ying Tung Graduate School. The authors thank Jingxiang Huang of Fok Ying Tung Graduate School for the assistance in OCEC analyzer maintenance. The authors are also grateful to Dr. Stephen M Griffith and Dr. Yongjie Li for the helpful comments. The authors gratefully acknowledge the NOAA Air Resources Laboratory (ARL) for the provision of the HYSPLIT transport and dispersion model used in this publication.

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

Table 1. Abbreviations.

| Abbreviation | Definition |
| --- | --- |
| $AAE_{470-660}$ | Absorption Angstrom Exponent between 470 and 660 nm |
| BB | Biomass burning |
| BrC | Brown Carbon |
| $D_{core}$, $D_{shell}$ | Particle diameter of core/shell |
| $E_{abs550}$ | Light absorption enhancement factor at 550 nm |
| $\sigma_{abs}550$ | Light absorption coefficient at 550 nm |
| $\sigma_{abs,t}$ | Total light absorption coefficient of a coated particle |
| $\sigma_{abs,p}$ | Primary light absorption coefficient attributed to the soot core alone of a coated particle |
| $\sigma_{abs,c}$ | Extra light absorption coefficient due to the lensing effect of coating on the soot core |
| LII | Laser induced incandescence technique for soot measurement |
| LWC | Liquid water content |
| $MAE_{550}$ | Mass absorption efficiency at 550 nm, also known as mass absorption cross-section (MAC) |
| $MAE_{p,550}$ | Primary MAE of freshly emitted soot particles at 550 nm |
| MAAP | Multi Angle Absorption Photometer |
| MOUDI | Micro Orifice Uniform Deposit Impactor |
| MRS | Minimum R squared method |
| PAS | Photo acoustic spectrometer |
| PRD | Pearl River Delta region, China |
| SP2 | Single particle soot photometer |
| SSA | Single scattering albedo |
| TD | Thermal denuder |
| TOA | Thermal optical analysis |
| TSV | Total spatial variance in backward trajectories cluster analysis |


Table 2. Comparison of MRS application on $(OC/EC)_p$ (for SOC estimation) and $MAE_p$ (for $E_{abs}$ estimation).

| | MRS in EC tracer method for SOC estimation (Wu and Yu, 2016) | MRS in EC tracer method for $E_{abs}$ estimation (this study) |
|---|---|---|
| **Key parameter of fresh EC particles to be determined** | $\left(\dfrac{OC}{EC}\right)_p = \dfrac{POC}{EC}$ | $MAE_p = \dfrac{\sigma_{abs,p}}{EC}$ |
| **Input quantities for MRS from measurements** | OC, EC (tracer) | $\sigma_{abs,t}$, EC (tracer) |
| **Variable to be decoupled by the tracer** | $OC = POC + SOC$ $= \left(\dfrac{OC}{EC}\right)_p \times EC + SOC$ | $\sigma_{abs,t} = \sigma_{abs,p} + \sigma_{abs,c}$ $= \dfrac{\sigma_{abs,p}}{EC} \times EC + \sigma_{abs,c}$ |
| **Ambient measurement at its closest to fresh emissions** | Minimum $R^2$ (SOC, EC) $SOC = OC - \left(\dfrac{OC}{EC}\right)_p \times EC$ | Minimum $R^2$ ($\sigma_{abs,c}$, EC) $\sigma_{abs,c} = \sigma_{abs,t} - MAE_p \times EC$ |
| **Graph** |  |  |


Table 3. Comparison of $E_{abs}$ between various studies.

| Location | Type | Sampling Duration | λ (nm) | Instrument | $E_{abs}$ | Method | Reference |
|---|---|---|---|---|---|---|---|
| Guangzhou, China | Suburban | 2012.2-2013.1 | 550 | AE+OCEC | 1.50±0.48 | MAE | This study |
| Xi'an, China | Urban | 2012.12-2013.1 | 870 | PAS | 1.8 | MAE | (Wang et al., 2014) |
| Shenzhen, China | Urban | 2011.8-9 | 532 | PAS | 1.3 | MAE | (Lan et al., 2013) |
| Jinan, China | Urban | 2014.2 | 678 | OCEC | 2.07 ± 0.72 | AFD | (Chen et al., 2017) |
| Nanjing, China | Suburban | 2012.11 | 532 | PAS | 1.6 | MAE | (Cui et al., 2016a) |
| Boulder, USA | Forest fire | 2010.9 | 532 | PAS | 1.38 | TD 200°C | (Lack et al., 2012a) |
| London, UK | Rural | 2012.2 | 781 | PAS | 1.4 | TD 250°C | (Liu et al., 2015) |
| California, USA | Rural | 2010.6 | 532 | PAS | 1.06 | TD 250°C | (Cappa et al., 2012) |
| Noto Peninsula, Japan | Rural | 2013.4-5 | 781 | PAS | 1.22 | TD 300°C | (Ueda et al., 2016) |
| Yuncheng, China | Rural | 2014.6-7 | 678 | OCEC | 2.25 ± 0.55 | AFD | (Cui et al., 2016b) |
| San Jose, Costa Rica | Rural | 2006 winter | 1064 | SP2 | 1.3 | Mie+SP2 | (Schwarz et al., 2008) |

AE: Aethalometer ; OCEC: OCEC analyzer; PAS: photo acoustic spectrometer; SP2: Single particle soot photometer; TD: Thermal
denuder AFD: filter filtration-dissolution

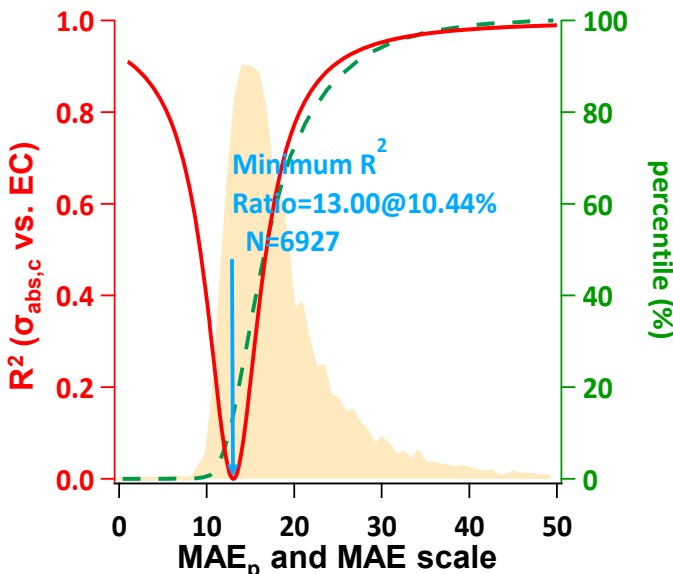


Figure 1. Minimum R squared (MRS) plot for calculating $MAE_p$ at 550 nm. The red curve is the correlation result between $\sigma_{abs,c}$ ($\sigma_{abs,t}$ − EC * $MAE_p$) and EC mass. The shaded area in light tan represents the frequency distribution of observed MAE. The dashed green line is the cumulative distribution of observed MAE.

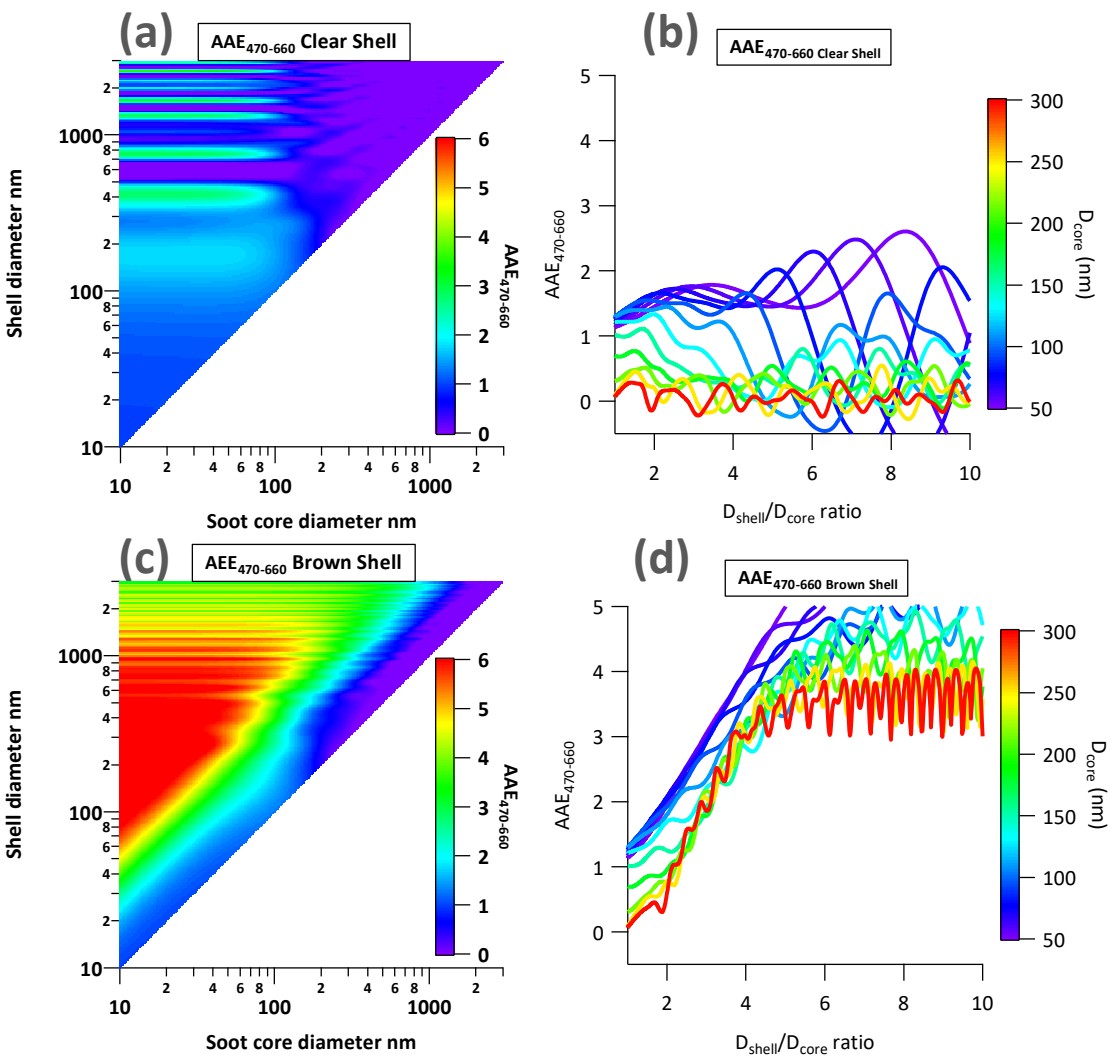


Figure 2. Mie simulated size dependency of soot particles $AAE_{470-660}$. (a) Combination of different clear shell (y axis) and
core diameters (x axis). The color coding represents the $AAE_{470-660}$ of a particle with specific core and clear shell size; (b)
Cross-sections views of (a). The color coding represents different $D_{core}$ in the range of 50 ~ 300 nm. (c)&(d) Similar to
(a)&(b) but from the brown shell scenario.



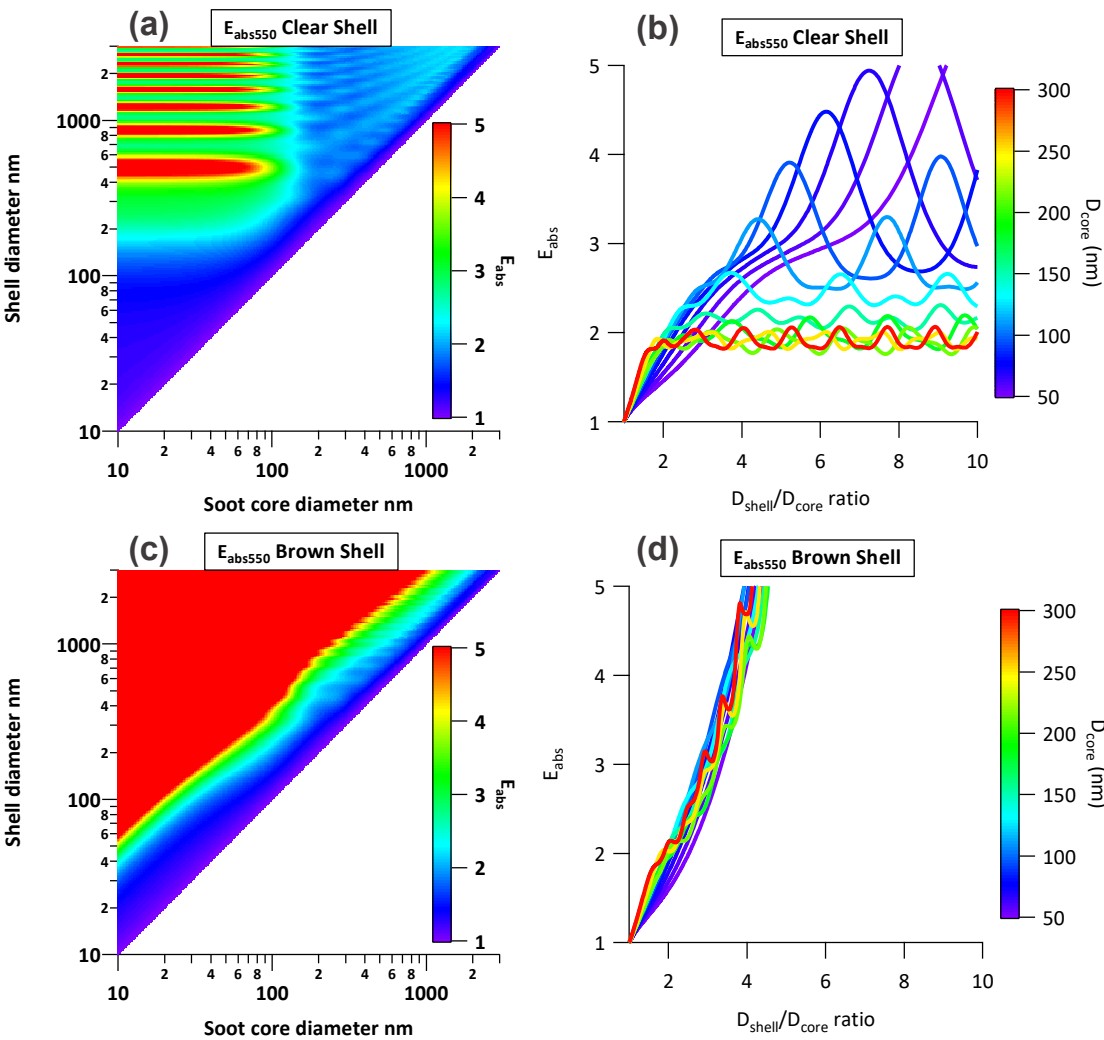

Figure 3. Mie simulated size dependency of soot particles $E_{abs}$ at wavelength 550 nm. (a) Combination of different clear shell (y axis) and core diameters (x axis). The color coding represents the $E_{abs}$ of a particle with specific core and clear shell size; (b) Cross-sections views of (a). The color coding represents different $D_{core}$ in the range of 50 – 300 nm. (c)&(d) Similar to (a)&(b) but from the brown shell scenario.

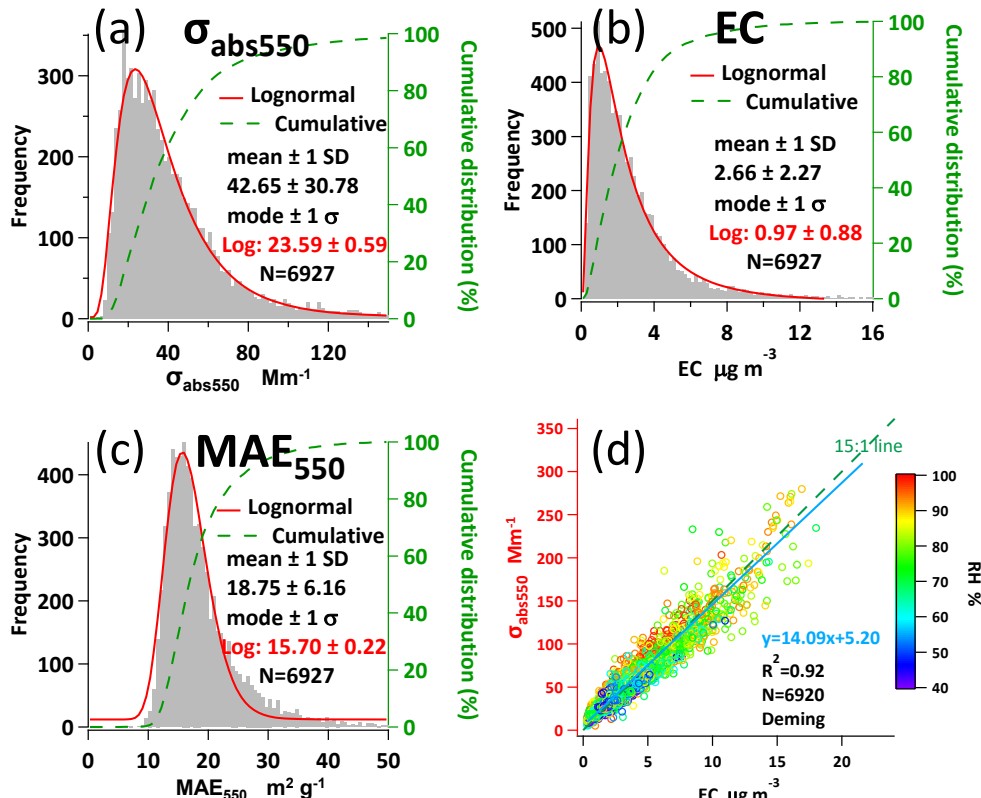

1096

Figure 4. Measured annual statistics of $\sigma_{abs550}$ , EC and MAE$_{550}$. (a) Annual frequency distribution of light absorption at
550 nm. The red curve represents the fitting line for a log-normal distribution. (b) Annual frequency distribution of EC
mass concentration (c) Frequency distribution of Mass absorption efficiency (MAE) at 550 nm. (d) Scatter plot of light
absorption (550 nm) and EC mass. The slope represents MAE$_{550}$. The blue regression line is by Deming regression. The
color coding represents RH.

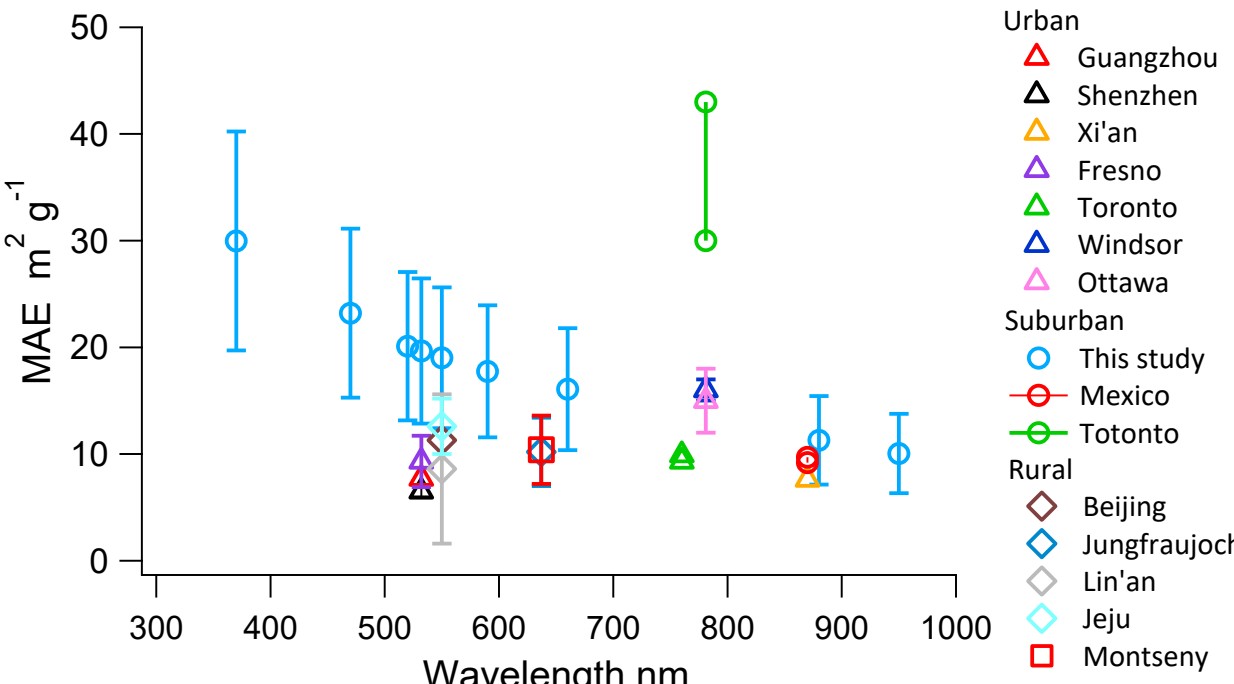


Figure 5. Comparison of spectral MAE measurements from this study with previous studies. Triangle, circle and rhombus
represent urban, suburban and rural respectively. Details and reference can be found in Table S1. The whiskers represent
one standard deviation.

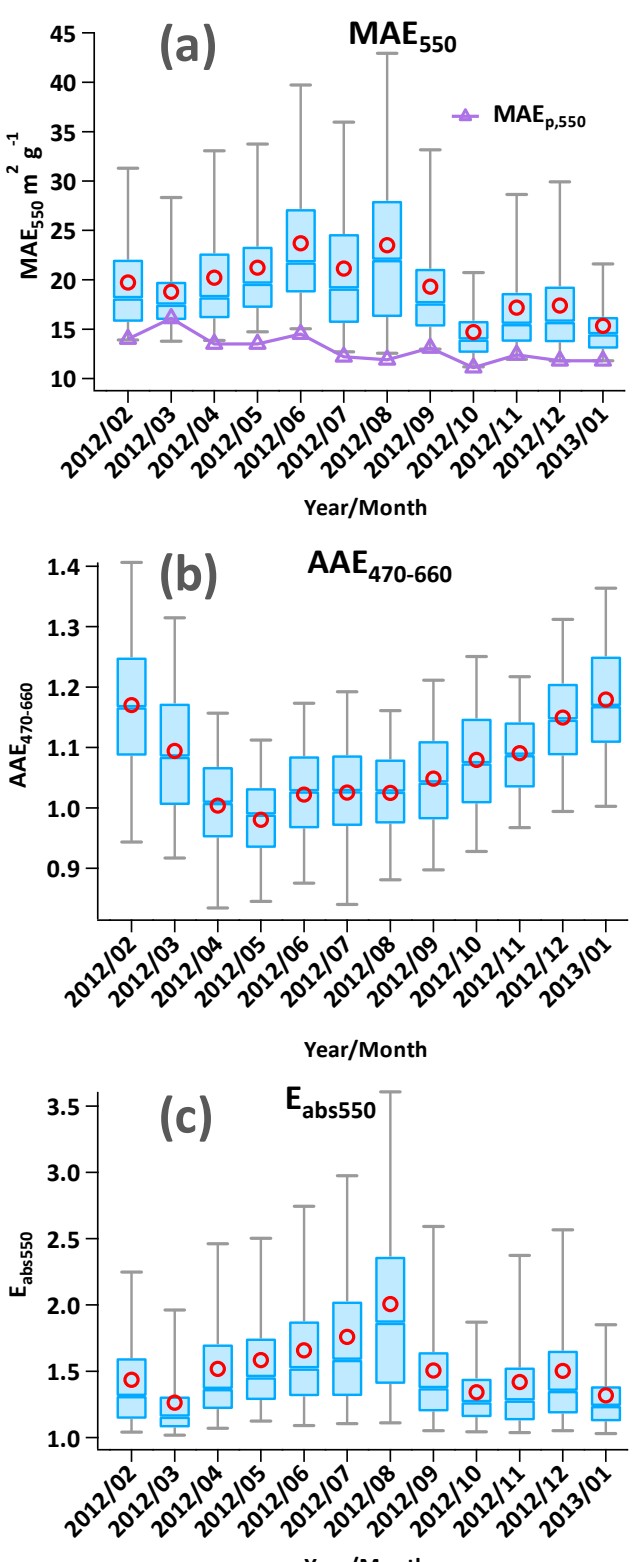


Figure 6. Measured monthly variations of (a) MAE$_{550}$, the purple line represents MAE$_{p,550}$ estimated by MRS (b) AAE$_{470-}$

$_{660}$ and (c) E$_{abs550}$. Red circles represent the monthly average. The line inside the box indicates the monthly median. Upper

and lower boundaries of the box represent the 75$^{th}$ and the 25$^{th}$ percentiles; the whiskers above and below each box represent

the 95$^{th}$ and 5$^{th}$ percentiles.

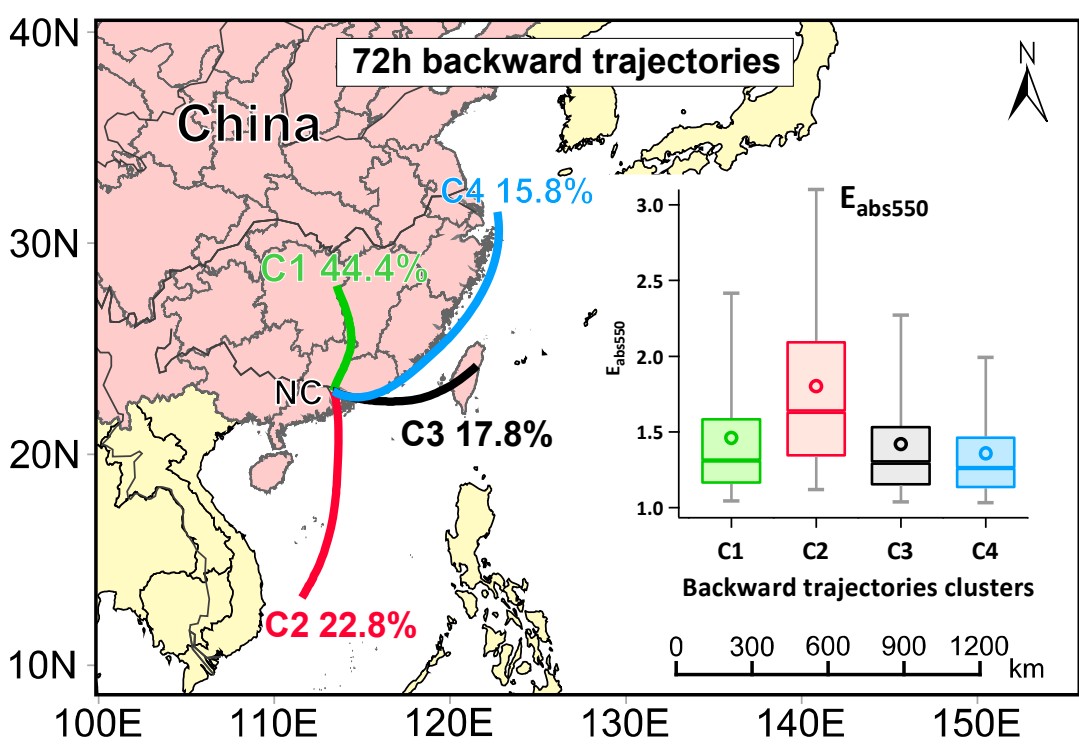

Figure 7. Average backward trajectories arriving at 100 m at NC site for four clusters (2012 Feb - 2013 Jan). $E_{abs550}$ by different clusters are shown in the box plot.

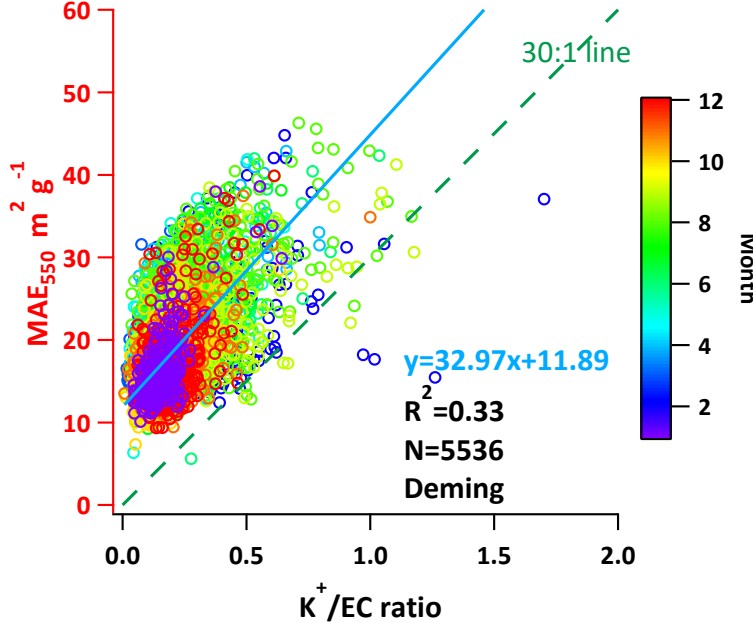

Figure 8. $MAE_{550}$ dependency on biomass burning indicator $K^+/EC$ ratio. The color coding represents months. The intercept represents MAE without biomass burning effect. The 30:1 line serves as a reference line with an integer slope that is close to the regressed slope through the origin.