# Peer review of "Quantifying black carbon light absorption enhancement by a novel"

_Atmospheric Chemistry and Physics, 2017_

## Referee Comment (RC1) · Anonymous Referee #2 · 17 Aug 2017

The manuscript presents a statistical analysis of black carbon light absorption enhancement based on observations made over roughly one year using a filter-based absorption instrument and a thermal-optical analysis OC/EC analyzer. They derive the absorption enhancement (Eabs) from the total mass absorption efficiency measured from the ratio of the absorption and EC measurements to that estimated for bare BC particles. To determine the bare BC MAE the authors employ a method that searches for an MAE value based on an assumed independence between EC and absorption due to BC coatings. While this approach presents a potential alternative to more expensive and labor-intensive methods to examine the important topic of BC light absorption enhancement, it is not clear to me that it works based on the information presented in the manuscript. First, while it does compare Eabs derived using this method to previ-

ously reported values using other techniques, there is not a direct comparison between the results obtained using this technique and more established methods for the same site. As pointed out by Liu et al. (2015), the "influence of coatings on BC absorption may be source and regionally specific", so it is difficult to draw too much confidence in the approach based on similarities with other locations. Second, it is not obvious that the measurement of EC is independent of the amount of light absorbed by BC coatings. While it is true that the mass of primary BC should be independent of coating absorption, EC is an operationally defined quantity, and depending on how coatings interact with its measurement during thermal optical analysis, could in fact have a relationship with coatings and light absorption due to coatings. For these two principle reasons I do not recommend the manuscripts publication in ACP in its current form.

General comments

RH impacts

The manuscript does not clearly state whether air was dried prior to sampling with the instruments. Section 4.5, which discusses impacts of RH on the observations, implies that it was not. The methods section states that 2.5 um cyclones were used upstream of the Aethalometer and Sunset instruments, but does not specify a sizecut for the nephelometer or MARGA instruments. If the nephelometer and MARGA instruments did not have a sizecut, it is extremely difficult to compare results from those instruments to those from the AE-31 and Sunset due to potential differences from coarse mode particle contributions.

If the air was not dried prior to sampling more details need to be provided regarding the effective RH for the optical instruments (AE-31 and nephelometer), which may be different from ambient RH due to temperature differences between the ambient air and the instruments. In addition, if air was sampled at near ambient RH it could have a number of complicating factors on the subsequent analysis. Filter-based absorption instruments, as the authors acknowledge, are affected by artifacts, including the scattering and shadowing of light in the filter matrix by filter fibers and particles embedded on the filter. Varying sample RH can change the filter artifacts in difficult to account for ways. For example, if hygroscopic particles are present on the filter they can swell or shrink depending on the sample RH, changing the light transmission properties of the filter, artifacts affecting the absorption measurement and possibly the absorption measurement itself depending on how RH interacts with the correction methods applied. The manuscript should include a discussion of how well the correction methods applied can account for changes arising from changes in RH. In addition, the effective size cut of the cyclone will be affected by RH in that the aerosol sampled will be different depending on how much water is associated with it. This effect may be small depending on the aerosol distribution and makeup of the light-absorbing particles, but should be addressed.

Another potential issue related to RH affects some of the results presented in Section 4.5. First, it is not clear if the RH reported is ambient RH or instrument RH. If ambient RH, it needs to be established that the RH at the location of the measurement (the filter) is also at the same RH as the ambient. In addition, it is not clear that the absorption measured on the AE-31 filter represents the absorption in the ambient air. For example, if BC becomes coated on the filter (which can happen in the presence of liquid organic aerosol (e.g. Subramanian et al., 2007) it is not clear how water uptake by those organic films might alter the absorption by particles that were not originally coated. Non-BC containing material can also take up water, alter the optical properties of the filter, and change the apparent absorption attributed to BC. Also, since the AE absorption measurement is based on the change in attenuation over time if the hygroscopic properties of particles change with time the apparent absorption could be affected. For example, consider a situation where BC with a hygroscopic coating has been sampled onto the filter for some period of time. As the sample RH begins to decrease the water will evaporate from the filter, likely leading to a decrease in attenuation (there is a reduced enhancement of absorption, for example). At the same time consider what happens if additional BC is sampled to the filter. This acts to increase

attenuation, opposing the RH effect. As a result the BC measured during this time would be underestimated (its effect on attenuation is countered to some degree by the opposing RH effect). Note that the magnitude of these impacts would be difficult to account for, in that they depend on the hygroscopicity of the material on the filter and how that material and associated water interacts in the filter matrix.

Measurement biases and impacts on the analysis

Biases in filter-based absorption and OC/EC measurements are mostly associated with other materials mixed and/or co-sampled with BC/EC, so it is likely that errors in the measurements have a systematic relationship. For example, the presence of organic aerosol and organic films has been linked to both biases in filter-based measurements (e.g., Lack et al., 2008) as well as potential impacts on EC measurement via pyrolized carbon correction (Subramanian et al., 2007). The manuscript needs a detailed discussion of how correlations in biases between the methods affect the retrieved MAEp values and resulting Eabs calculations. Since the manuscript is seeking to establish a new method this potential issue needs much more attention. Several specific comments below are related to this general concern.

Specific comments

78-80: Missing from the list of field studies is Cappa et al. (2012), which found a weak enhancement for observations in California, US.

85: Suggest changing to "The TD approach is briefly discussed here."

90: Some good points are raised in this discussion, but a few things are lacking. First, another major reason for use of PAS systems in the approach is that the technique does not have artifacts associated with filter-based methods for measuring light absorption, thus it provides an unambiguous measure of the light absorption coefficient in both heated and un-heated states. Also missing from the discussion is potential differences in BC core morphology being different for "fresh" versus "aged with coating removed"

conditions, or in other words, the TD may not be a perfect "time machine" to reverse the aging process and determine the effects of BC aging.

100-101: I realize this is not the focus of the manuscript, but it is worth noting here the logic presented here assumes a linear relationship between MAE and coated soot particle number fraction measured by SP2. This may not be the case due to limitations in the size of BC particles measured by the SP2, its ability to measure thin coatings, the potential impact of thin coatings on MA, the assumption of core-shell morphology, and the nature of the relationship between the fraction of coated particles (a parameter which ignores coating thickness, the physical driver of the absorption enhancement) and MAE.

133: What does the 5% uncertainty refer to? Is this before or after corrections? Are the uncertainties in the corrections as low as 5%? Also, the manuscript gives an uncertainty of 24% for the ECOC analyzer here, but then later states (line 153) up to a factor of 5 differences in EC measured by different protocols are possible. How is this to be reconciled?

166-169: The argument given here is only true if the errors in absorption measurement and EC mass measurement are independent and random. If some factor causes a positive bias in the filter-based absorption but a negative bias in the EC measurement they will not cancel, but instead there will be an apparent and potentially false apparent absorption enhancement. Or consider a situation where some co-sampled material affects the OC/EC split but not the filter-based absorption measurement, leading to an apparent change in Eabs.

187-188: Since EC is an operationally defined parameter based on a measurement technique there is not necessarily an inherent interdependency between it and the absorption due to coatings gained following emission.

Section 3.2: The results from the Mie simulations are not a new contribution and main points drawn from the discussion could be drawn from previous studies given in the

literature (e.g., Lack and Cappa, 2010; Bond et al., 2006).

260: Should note here that the AAE observed in measurement studies also includes absorption by potentially externally mixed brown carbon particles, not just those present in the form of shells on BC particles.

295: Suggest changing "concentrations" to "modes".

299-302: Minor point here, but the MOUDI data also reflect differences in BC core equivalent diameters measured using LII and aerodynamic diameter of BC, so we would not necessarily expect similar sizing, even for uncoated BC. Worth mentioning here, though I do not disagree with the main argument given here.

303: Please state whether Tan et al. refer to BC only diameter or mixed particle diameter here.

314: I am curious as to why the 470 and 660 nm wavelength pairs were used to quantify AAE. Brown carbon tends to show much stronger impact at shorter wavelengths. Could the authors please include the AAE determined for the UV and 880 nm channels also measured by the AE-31?

321: I assume MAEp should be MAEp, 550 based on Table 1?

325-328: The Cappa et al. observation showing weak enhancement should also be included in this summary.

355: Use of "significantly" implies a statistically significant difference between the clusters. Please provide uncertainty estimates and confidence levels if this is intended, or omit.

360-361: I am curious how much the air mass trajectories were influenced by precipitation during the monsoon period/rainy season. It seems like aged/coated BC likely having higher Eabs would be more susceptible to removal compared to less-aged/coated BC with lower Eabs?

375-376: Could the authors clarify the reasoning given here? First, I believe the authors mean the MAEp determined in section 4.1, not 4.3, correct? I think this is arguing that because MAEp is, on average, higher than the MAE observed in the absence of BB influence there must be a large amount of BB influence on BC at this location? If so, I think some caution or caveats should be included in the discussion, since the BB influence tracer is not independent of the EC measurement. Further, I would expect there to be a much stronger effect on AAE if BB was such an important BC source, yet the later sections establish that AAE does not show a response.

381: Again I am curious as to the reasoning for selection of these wavelength pairs.

384-386: I do not quite follow the reasoning presented here. Why does the monthly average of 1.2 suggest variations in AAE are associated with thicker coatings than BrC contribution? To me the strongest evidence of minimal BB contribution is the lack of correlation with the K+/EC tracer. Does AAE show any correlation with Eabs? The manuscript asserts AAE is dominated by coating, so there should be a relationship based on the arguments presented.

413-415: The refractive index of the shell also changes as it takes up water. Is this accounted for in the modeling?

426: This approach requires that only BC and associated coatings affect Eabs and AAE, but AAE can also be affected by non-BC aerosol (e.g., brown carbon or dust) that is externally mixed with BC. I am not sure how this approach can work unless there is clear evidence that there are no other light-absorbing particles or that the relative abundance does not change with season.

Table 1, while convenient, could be omitted for length.

Figures 2-4 do not add much to the manuscript, in my opinion, and could be removed for length.

Figure 7: Minor, but x-axis is year and month, not just month. Replacing with "Date"

would be fine.

References:

Bond et al. (2006) Limitations in the enhancement of visible light absorption due to mixing state, J. Geophysical Research, doi: 10.1029/2006JD007315

Lack and Cappa (2010) Impact of brown and clear carbon on light absorption enhancement, single scatter albedo and absorption wavelength dependence of black carbon, Atmospheric Chemistry and Physics, doi: 10.5194/acp-10-4207-2010

Lack et al. (2008) Bias in filter-based light absorption measurements due to organic aerosol loading: Evidence from ambient measurements, Aerosol Science and Technology, 42, doi: 10.1080/02786820802389277

Liu et al. (2015) Enhanced light absorption by mixed source black and brown carbon particles in UK winter, Nature Communications, doi: 10.1038/ncomms9435

Subramanian et al. (2007) Yellow beads and missing particles: Trouble ahead for filter-based absorption measurements, Aerosol Science and Technology, doi: 10.1080/02786820701344589

---

## Referee Comment (RC2) · Anonymous Referee #1 · 1 Sep 2017

This paper proposed a new approach to characterize the light absorption enhancement of BC and reported its application to the observed results based on an aethalometer and a thermo-optical carbon analyzer in Pearl River Delta area. This manuscript includes sufficient originality, and the topic seems to fit the journal. However, there are several concerns on the accuracy of the light absorption measurements, definition of Eabs, and discussions on the observed optical properties. I believe that the points below should be addressed before considering the publication in ACP.

General comments

1) Accuracy of the aethalometer measurements Although the authors used the correction scheme reported by Weingartner et al. (2003) to correct the artifacts due to aerosol loading, filter matrix and scattering effect, it is not clear that the light absorption

for "coated-BC" can be accurately measured (or corrected), especially under "high RH" conditions. For example, Arnott et al (2003) reported that a filter-based photometer has large potential artifacts above 80% RH. The authors need to give more detailed information on accuracy of aethalometer measurements.

2) Definitions of the MAEp and Eabs The MAEp obtained by the method in this study is MAE value for primary emission source. I think that the MAEp, which was obtained in this study, includes the effects of "lensing effect" due to co-emitted OC. Therefore, the definition of Eabs values should be different with the Eabs in the previous studies, at least, using TD technique. Detailed explanation on this point should be added.

3) The effect of biomass burning

3-1) Lines 371-374: "During the rainy season when oceanic prevailing wind dominates, BC from BB emission in Southeast Asia can reach PRD through long range transport (LRT), resulting in an elevated K+/EC ratio and MAE550, which might be a combination of a thicker coating when freshly emitted from BB sources and enhanced coating during LRT."

=> If so, I think the MAEp values should be higher in summer (rainy season). Please consider about this point.

3-2) Lines 384-388: "Since the monthly average AAE in wintertime didn't exceed 1.2 (Table S3), the variations of AAE in the PRD are likely more associated with thicker coatings rather than the contribution of BrC. The results also imply that attempts on BrC absorption attribution for the PRD dataset presented in this study could be risky, considering that elevation of AAE is actually dominated by coating (Lack and Langridge, 2013)."

=> Because higher Eabs values were observe in summer, the coating would be thicker in summer. The suggestion that "the higher AAE in winter are likely more associated with thicker coatings" is not reasonable. Please consider about this point.

4) The effect of relative humidity (RH) on optical properties As mentioned above, the authors need to show some evidences suggesting that the obtained positive correlation between f(RH) (i.e., MAE) and RH is not due to the artifact of RH on aethalometer measurements. If the measurements are assumed to be accurate, the observed higher Eabs (and MAE) and lower AAE values in summer may be explained only by RH, considering that the RH may be higher in summer and RH for air masses from Southeast Asia (C2). Which one of higher OC/EC ratio and higher RH do you suggest as a main contributor for observed higher Eabs in summer?

Specific comments

1) Line 235-236 "Thus, a BrC coating (brown shell) scenario is also considered in Mie simulation following the wavelength dependent RI suggested by Lack and Cappa (2010)." => Because the results of Mie simulation should be varied depending on the RI values given here, I recommend adding the range of RI values.

2) Section 3.2.4, 1st paragraph For the brown shell scenario, I think that both of lensing effect and light absorption by BrC contribute to the Eabs values. I recommend adding the fraction of each contribution.

3) Section 4.1, 2nd paragraph The annual average MAEp value (13 m2/g) is larger than those estimated for bare BC. It may indicate that the obtained MAEp values include the effects of "lensing effect" due to co-emitted OC, as mentioned above. If so, caution on the definition of MAEp in each study should be taken during comparison with other studies. Also, it is better to add information on the wavelength used in each study.

4) Section 4.5 How did the absorption coefficients for dry conditions determine to obtained f(RH)?

5) Figure 9 What do to want to suggest from 30:1 line in Fig. 9?

(References) Arnott, W. P., Moosmuller, H., Sheridan, P. J., Ogren, J. A., Raspet, R.Slaton, W. V. 2003. Photoacoustic and Filter-Based Ambient Aerosol Light Absorption Measurements: Instrument Comparisons and the Role of Relative Humidity. J. Geophys. Res.-Atmos., 108: 4034–4045.

---

## Author Comment (AC1) · 15 Oct 2017

**Point-by-point response to review comments on manuscript acp-2017-582 "Quantifying black carbon light absorption enhancement by a novel statistical approach"**

**By Cheng Wu et al.**

We thank the two anonymous reviewers for their constructive comments to improve the manuscript. Our point-by-point responses to the review comments are listed below. Changes to the manuscript are marked in blue in the revised manuscript. The marked manuscript is submitted together with this response document.

**Anonymous Referee #2**

**R2-Q1.** The manuscript presents a statistical analysis of black carbon light absorption enhancement based on observations made over roughly one year using a filter-based absorption instrument and a thermal-optical analysis OC/EC analyzer. They derive the absorption enhancement ($E_{abs}$) from the total mass absorption efficiency measured from the ratio of the absorption and EC measurements to that estimated for bare BC particles. To determine the bare BC MAE the authors employ a method that searches for an MAE value based on an assumed independence between EC and absorption due to BC coatings. While this approach presents a potential alternative to more expensive and labor-intensive methods to examine the important topic of BC light absorption enhancement, it is not clear to me that it works based on the information presented in the manuscript. First, while it does compare $E_{abs}$ derived using this method to previously reported values using other techniques, there is not a direct comparison between the results obtained using this technique and more established methods for the same site. As pointed out by Liu et al. (2015), the "influence of coatings on BC absorption may be source and regionally specific", so it is difficult to draw too much confidence in the approach based on similarities with other locations. Second, it is not obvious that the measurement of EC is independent of the amount of light absorbed by BC coatings. While it is true that the mass of primary BC should be independent of coating absorption, EC is an operationally defined quantity, and depending on how coatings interact with its measurement during thermal optical analysis, could in fact have a relationship with coatings and light absorption due to coatings. For these two principle reasons I do not recommend the manuscripts publication in ACP in its current form.

**Author's Response:**

(1) A comparison would definitely be helpful for the verification of the MRS approach in the future. However, as the reviewer has pointed out, TD approach may not be a prefect "time machine" to reverse the aging process for $E_{abs}$ determination. The definition of $MAE_p$ by MRS is different from the MAEp by TD. $MAE_p$ by MRS reflect the MAE of soot particles at the emission source, while $MAE_p$ by TD reflect the MAE of bare soot particles. First, the morphology of thermally denuded soot particles (collapsed globules) is different from the morphology of freshly emitted BC particles (fractal-like aggregates). Second, most of the coatings are removed for TD denuded soot particles, but freshly emitted soot particles usually come with a thin coating due to the temperature drop from engine to the ambient air. As a result, the $MAE_p$ by TD approach is expected to be lower than the $MAE_p$ of emission source retrieved by MRS. In this sense, it may be difficult to extract quantitative information through such comparisons due to the difference in $MAE_p$ definition.

(2) We agree with the reviewer that coating can affect EC determination by the thermal-optical analysis. Lee et al. (2007) used artificially fabricated EC samples with OC coatings to evaluate the performance of OC/EC analysis. Biases were observed but the results are linearly correlated ($R^2>0.9$) with the true OC and EC values. As shown in Figure R-5 and discussions in the following response to R2-Q5, the $E_{abs}$ estimation by MRS is insensitive to systematic biases. The MRS approach mainly rely on correlation analysis, the systematic biases are not a big concern as long as the measured data correlated well with the true values.

Please see below for point-by-point response to reviewers' comments.

**General Comments**

**R2-Q2.** RH impacts

The manuscript does not clearly state whether air was dried prior to sampling with the instruments. Section 4.5, which discusses impacts of RH on the observations, implies that it was not. The methods section states that 2.5 um cyclones were used upstream of the Aethalometer and Sunset instruments, but does not specify a size cut for the nephelometer or MARGA instruments. If the nephelometer and MARGA instruments did not have a size cut, it is extremely difficult to compare results from those instruments to those from the AE-31 and Sunset due to potential differences from coarse mode particle contributions.

**Author's Response:**
Both nephelometer and MARGA are equipped with a $PM_{2.5}$ inlet to remove the coarse particles. The following text is added in section 2 to improve the clarity.

Both instruments are equipped with a $PM_{2.5}$ inlet to remove the coarse particles.

**R2-Q3.** If the air was not dried prior to sampling more details need to be provided regarding the effective RH for the optical instruments (AE-31 and nephelometer), which may be different from ambient RH due to temperature differences between the ambient air and the instruments. In addition, if air was sampled at near ambient RH it could have a number of complicating factors on the subsequent analysis. Filter-based absorption instruments, as the authors acknowledge, are affected by artifacts, including the scattering and shadowing of light in the filter matrix by filter fibers and particles embedded on the filter. Varying sample RH can change the filter artifacts in difficult to account for ways. For example, if hygroscopic particles are present on the filter they can swell or shrink depending on the sample RH, changing the light transmission properties of the filter, artifacts affecting the absorption measurement and possibly the absorption measurement itself depending on how RH interacts with the correction methods applied. The manuscript should include a discussion of how well the correction methods applied can account for changes arising from changes in RH. In addition, the effective size cut of the cyclone will be affected by RH in that the aerosol sampled will be different depending on how much water is associated with it. This effect may be small depending on the aerosol distribution and makeup of the light-absorbing particles, but should be addressed.

**Author's Response:** Thanks for the very insightful comments. We fully agree that RH could be a source of uncertainty for filter based $\sigma_{abs}$ measurement. Arnott et al. (2003) found particle soot absorption photometer (PSAP) showed erratic response as RH change. Schmid et al. (2006) observed dependency of PSAP $\sigma_{abs}$ on RH, but the effect of RH on Aethalometer performance is negligible. Inter-comparison studies shown that with proper corrections, Aethalometer $\sigma_{abs}$ agrees well with PAS (Ajtai et al., 2011). During the inter-comparison study of Aethalometer (AE-16) and PAS in Guangzhou (Wu et al., 2009), a good correlation was found ($R^2$=0.96) as shown in Figure R1 or Figure S1. These filed comparison results imply that Aethalometer results are linearly correlated with PAS and RH has limited effect on Aethalometer measurements. Please refer to our response to R1-Q2 for the revisions made in the manuscript.

**R2-Q4.** Another potential issue related to RH affects some of the results presented in Section4.5. First, it is not clear if the RH reported is ambient RH or instrument RH. If ambient RH, it needs to be established that the RH at the location of the measurement (the filter) is also at the same RH as the ambient. In addition, it is not clear that the absorption measured on the AE-31 filter represents the absorption in the ambient air. For example, if BC becomes coated on the filter (which can happen in the presence of liquid organic aerosol (e.g. Subramanian et al., 2007) it is not clear how water uptake by those organic films might alter the absorption by particles that were not originally coated. Non-BC containing material can also take up water, alter the optical properties of the filter, and change the apparent absorption attributed to BC. Also, since the AE absorption measurement is based on the change in attenuation over time if the hygroscopic properties of particles change with time the apparent absorption could be affected. For example, consider a situation where BC with a hygroscopic coating has been sampled onto the filter for some period of time. As the sample RH begins to decrease the water will evaporate from the filter, likely leading to a decrease in attenuation (there is a reduced enhancement of absorption, for example). At the same time consider what happens if additional BC is sampled to the filter. This acts to increase attenuation, opposing the RH effect. As a result the BC measured during this time would be underestimated (its effect on attenuation is countered to some degree by the opposing RH effect). Note that the magnitude of these impacts would be difficult to account for, in that they depend on the hygroscopicity of the material on the filter and how that material and associated water interacts in the filter matrix.

**Author's Response:** The RH used in this study is ambient RH. We agree with the reviewer that RH at the filter of Aethalometer is different from the ambient RH. The RH in the optical chamber of Aethalometer may be lower than the ambient RH due to the slightly elevated temperature inside the instrument. The magnitude of RH difference is similar between different instruments: 20% for the Aethalometer (Schmid et al., 2006) and 15% for the nephelometer (Guyon et al., 2004). Although the RH in the Aethalometer optical chamber was not measured in this study, its level was expected to be slightly lower than the ambient RH. Schmid et al. (2006) found PSAP $\sigma_{abs}$ showed dependency on RH, but the effect of RH on Aethalometer performance was negligible. Cappa et al. (2008) found $\sigma_{abs}$ by PSAP maintained high linearity with PAS even under high RH conditions (65-91%). Inter-comparison studies shown that with appropriated corrections, Aethalometer $\sigma_{abs}$ agrees well with PAS (Ajtai et al., 2011). During the inter-comparison study of Aethalometer (AE-16) and PAS in Guangzhou (Wu et al., 2009), good correlation was found ($R^2$=0.96) as shown in Figure S1. These comparison results imply that Aethalometer results are linearly correlated with PAS and RH has limited interference on Aethalometer measurements.

**R2-Q5.** Measurement biases and impacts on the analysis Biases in filter-based absorption and OC/EC measurements are mostly associated with other materials mixed and/or co-sampled with BC/EC, so it is likely that errors in the measurements have a systematic relationship. For example, the presence of organic aerosol and organic films has been linked to both biases in filter-based measurements (e.g., Lack et al., 2008) as well as potential impacts on EC measurement via pyrolized carbon correction (Subramanian et al., 2007). The manuscript needs a detailed discussion of how correlations in biases between the methods affect the retrieved $MAE_p$ values and resulting $E_{abs}$ calculations. Since the manuscript is seeking to establish a new method this potential issue needs much more attention. Several specific comments below are related to this general concern.

**Author's Response:** The bead-like particles on filter fibers could interfere both Aethalometer and OC/EC measurements. Since the study by Subramanian et al. (2007) was based on source samples from low temperature BB emissions, how this bead-like particles could affect ambient measurements depends on their fractional contribution to $PM_{2.5}$. Cappa et al. (2008) found that the light absorption enhancement factor for the PSAP relative to the PAS due to the presence of externally mixed organic aerosols is proportional to the OA-to-soot ratio, which means this bias is systematic. We conduct two tests to investigate the effect of systematic EC and $\sigma_{abs}$ bias on $E_{abs}$ estimation. As shown in Figure R-5, the MRS approach is insensitive to systematic bias in EC and $\sigma_{abs}$ measurements. Details of the tests are added in section 4.1 and also shown below.

To investigate the performance of the MRS approach in response to systematic bias in EC and $\sigma_{abs}$, two simple tests are conducted as shown in Figures S9 and S10 by adding systematic biases to $\sigma_{abs550}$ and EC. Test A represents a situation when $\sigma_{abs}$ is overestimated and EC is underestimated. The biased data are marked as $\sigma'_{abs550}$ and EC' respectively, as shown below:

$$\sigma'_{abs550} = \sigma_{abs550} \times 2 \qquad (8)$$
$$EC' = EC \times 0.7 \qquad (9)$$

As a result, the average $MAE_{550}$ changed from 18.75 to 53.58 $m^2\,g^{-1}$ and $MAE_p$ changed from 13 to 37 $m^2\,g^{-1}$ (Figure S9). However, $E_{abs}$ by ratio of averages remain the same (1.44).

In Test B, EC by different TOA protocols are compared to investigate the effect of different EC determination approaches while $\sigma_{abs550}$ remains unchanged. EC by IMPROVE TOR protocol is calculated from NIOSH TOT EC following an empirical formula for suburban sites derived from a 3-year OCEC dataset in PRD (Wu et al., 2016):

$$EC_{IMP\_TOR} = 2.63 \times EC_{NSH\_TOT} + 0.05 \tag{10}$$

As shown in Figure S10, $MAE_{550}$ changed from 18.75 to 7.02 $m^2\,g^{-1}$ and $MAE_p$ changed from 13 to 5 $m^2\,g^{-1}$, but $E_{abs}$ remain almost the same (1.40). Result of Test B implies that although EC is operationally defined, the discrepancy of EC between TOA protocols did not weaken the role of EC serving as a tracer for primary emissions in MRS application. These examples demonstrate that systematic biases in $\sigma_{abs550}$ and EC have no effects on $E_{abs}$ estimation by the MRS approach.

[Figure]

**Figure R-5.** Comparison of $E_{abs}$ from original data and systematically biased data (Test A).

**Specific Comments**

**R2-Q6.** 78-80: Missing from the list of field studies is Cappa et al. (2012), which found a weak enhancement for observations in California, US.

**Author's Response:** Suggestion taken. Cappa et al. (2012) was added.

**R2-Q7.** 85: Suggest changing to "The TD approach is briefly discussed here."

**Author's Response:** Revision made.

**R2-Q8.** 90: Some good points are raised in this discussion, but a few things are lacking. First, another major reason for use of PAS systems in the approach is that the technique does not have artifacts associated with filter-based methods for measuring light absorption, thus it provides an unambiguous measure of the light absorption coefficient in both heated and un-heated states. Also missing from the discussion is potential differences in BC core morphology being different for "fresh" versus "aged with coating removed" conditions, or in other words, the TD may not be a perfect "time machine" to reverse the aging process and determine the effects of BC aging.

**Author's Response:** Thanks for the suggestions. The following content is added in the main text.

As an in-situ technique, PAS eliminates the artifacts associated with filter-based methods (Weingartner et al., 2003;Coen et al., 2010) and is often considered as the reference instrument for light absorption coefficient determination (Arnott et al., 2003;Arnott et al., 2005).

It's also worth noting that $MAE_p$ by the TD approach is different from the $MAE_p$ at the emission source. First, the morphology of thermally denuded BC particles (compact aggregates) is different from that of freshly emitted BC particles (chain-like aggregates). Second, most of the coatings is removed with the TD denuded soot particles, but freshly emitted soot particles usually come with a thin coating of OC formed from condensation of OC vapors as the temperature drops from engine to the ambient air. As a result, the $MAE_p$ by TD approach is expected to be lower than the $MAE_p$ of emission source. In this sense, the TD approach may not be a prefect "time machine" to reverse the aging process for $E_{abs}$ determination.

**R2-Q9.** 100-101: I realize this is not the focus of the manuscript, but it is worth noting here the logic presented here assumes a linear relationship between MAE and coated soot particle number fraction measured by SP2. This may not be the case due to limitations in the size of BC particles measured by the SP2, its ability to measure thin coatings, the potential impact of thin coatings on MA, the assumption of core-shell morphology, and the nature of the relationship between the fraction of coated particles (a parameter which ignores coating thickness, the physical driver of the absorption enhancement) and MAE.

**Author's Response:** The following text is added in the main text for clarification.

It is worth noting that this approach provides only a rough approximation of $E_{abs}$ since the parameter used here (coated soot particles number fraction) ignores other main drivers of light absorption enhancement (e.g. coating thickness). As a result, this approach is only valid for a period of measurements, for which coating thickness is relatively constant and the MAE variations are dominated by coated soot particles number fraction.

**R2-Q10.** 133: What does the 5% uncertainty refer to? Is this before or after corrections? Are the uncertainties in the corrections as low as 5%? Also, the manuscript gives an uncertainty of 24% for the ECOC analyzer here, but then later states (line 153) up to a factor of 5 differences in EC measured by different protocols are possible. How is this to be reconciled?

**Author's Response:** The 5% measurement uncertainty refers to instrument precision. An uncertainty of 24% for the ECOC analyzer is the instrument precision when NIOSH protocol is applied for TOA. The 5 times differences in EC measured by different protocols arises from inter-protocol discrepancy. It's a concept different from the instrument precision. To avoid confusion, we change the term "measurement uncertainty" to "measurement precision" throughout the manuscript.

**R2-Q11.** 166-169: The argument given here is only true if the errors in absorption measurement and EC mass measurement are independent and random. If some factor causes a positive bias in the filter-based absorption but a negative bias in the EC measurement they will not cancel, but instead there will be an apparent and potentially false apparent absorption enhancement. Or consider a situation where some co-sampled material affects the OC/EC split but not the filter-based absorption measurement, leading to an apparent change in $E_{abs}$.

**Author's Response:** We conduct two tests to investigate the effect of systematic EC and $\sigma_{abs}$ bias on $E_{abs}$ estimation. As shown in Figure R-5 and discussions in the response to R2-Q5, the MRS approach is insensitive to systematic bias in EC and $\sigma_{abs}$ measurements.

Following contents are added in the manuscript.

**In section 2.1**

Systematic bias in MAE (e.g. overestimation of $\sigma_{abs}$ and variability of EC mass by different TOA protocols) discussed above have little effect on $E_{abs}$ estimation by MRS. As shown in Eq. 3, $E_{abs}$ is the ratio of $MAE_t$ to $MAE_p$ or $\sigma_{abs,t}$ to $\sigma_{abs,p}$, thus most of the bias in EC mass or $\sigma_{abs}$ is cancelled out during the $E_{abs}$ calculation. More details are discussed in section 4.1.

Please refer to our response to R2-Q5 for contents added in section 4.1.

**R2-Q12.** 187-188: Since EC is an operationally defined parameter based on a measurement technique there is not necessarily an inherent interdependency between it and the absorption due to coatings gained following emission.

**Author's Response:** It is true that EC is operationally defined. But studies have shown that EC by different protocols correlate very well (Chow et al., 2001;Chow et al., 2004;Wu et al., 2012;Wu et al., 2016). The discrepancy of EC between thermal-optical analysis protocols did not weaken the role of EC to serve as a tracer for primary emission in MRS application, as shown in the Test B in the response to R2-Q5. On the other hand, extra absorption due to coating is associate with secondary process after emission. The variations of primary emission are relatively independent to the variations of secondary process.

**R2-Q13.** Section 3.2: The results from the Mie simulations are not a new contribution and main points drawn from the discussion could be drawn from previous studies given in the literature (e.g., Lack and Cappa, 2010; Bond et al., 2006).

**Author's Response:** The main purpose of including Mie simulations in this study is to help readers to understand the viability of $E_{abs}$ and AAE form a theoretical perspective and their dependence on different core/shell diameter combinations. Citing main points from literature is one way but that could be distractive. From a reader point of view, digging the detail information from the literature might not be as convenient as reading the Mie simulations in this study. In addition, Mie simulations figures shown here are specifically plotted to support the later discussion of the measurement results. We feel that the inclusion of Mie simulations provides a smooth one-stop reading experience.

**R2-Q14.** 260: Should note here that the AAE observed in measurement studies also includes absorption by potentially externally mixed brown carbon particles, not just those present in the form of shells on BC particles.

**Author's Response:** Suggestion taken. The corresponding content has been revised as follows.

These high AAE results are consistent with the previous model study (Lack and Cappa, 2010) and could partially explain the high AAE observed in measurement studies (Kirchstetter et al., 2004;Hoffer et al., 2006), since the presence of externally mixed BrC particles also contribute to the wavelength dependent light absorption.

**R2-Q15.** 295: Suggest changing "concentrations" to "modes".

**Author's Response:** Revision applied.

**R2-Q16.** 299-302: Minor point here, but the MOUDI data also reflect differences in BC core equivalent diameters measured using LII and aerodynamic diameter of BC, so we would not necessarily expect similar sizing, even for uncoated BC. Worth mentioning here, though I do not disagree with the main argument given here.

**Author's Response:** Thanks for the clarification. The following content is added to remind readers the difference when comparing the sizing between LII and MOUDI.

BC sizing by LII is based on volume equivalent diameter (VED), while MOUDI is based on aerodynamic diameter. As a result, these two techniques do not necessarily yield similar sizes, even for the bare soot particles. The conversion between these two types of diameters involves the knowledge of particle density and morphology (drag force).

**R2-Q17.** 303: Please state whether Tan et al. refer to BC only diameter or mixed particle diameter here.

**Author's Response:** Tan et al. refers to coated BC diameter. The corresponding content has been revised as follows to improve the clarity.

A recent closure study on BC mixing state in the PRD region suggests $\sigma_{abs}$ is dominated by coated soot particles in the range of 300~400 nm (Tan et al., 2016).

**R2-Q18.** 314: I am curious as to why the 470 and 660 nm wavelength pairs were used to quantify AAE. Brown carbon tends to show much stronger impact at shorter wavelengths. Could the authors please include the AAE determined for the UV and 880 nm channels also measured by the AE-31?

**Author's Response:** The 470 and 660 nm wavelength pairs were used to represent AAE in the visible range. $AAE_{370-880}$ (1.13±0.13) is added as shown in Figure S8b, which is slightly higher than $AAE_{470-660}$ (1.09±0.13) shown in FigureS8a.

**R2-Q19.** 321: I assume $MAE_p$ should be $MAE_{p,550}$ based on Table 1?

**Author's Response:** Thanks for pointing out. Revision made.

**R2-Q20.** 325-328: The Cappa et al. observation showing weak enhancement should also be included in this summary.

**Author's Response:** Suggestion taken. Cappa et al. (2012) was added.

**R2-Q21.** 355: Use of "significantly" implies a statistically significant difference between the clusters. Please provide uncertainty estimates and confidence levels if this is intended, or omit.

**Author's Response:** Results of Wilcoxon-Mann-Whitney tests are included in SI to support the statement and also shown below. Wilcoxon-Mann-Whitney tests between C1&C2, C2&C3 and C2&C4 all show P<0.01, indicating that the mean of C2 is significantly higher than C1, C3 and C4.

[Figure]

Figure R-6 Frequency distributions of $E_{abs550}$ by different air mass clusters.

**R2-Q22.** 360-361: I am curious how much the air mass trajectories were influenced by precipitation during the monsoon period/rainy season. It seems like aged/coated BC likely having higher $E_{abs}$ would be more susceptible to removal compared to less-aged/coated BC with lower $E_{abs}$?

**Author's Response:** It's difficult to directly evaluate the rain effect on $E_{abs550}$ for corresponding air mass trajectories since the measurement is only conducted at the end point of air mass trajectories. Alternatively, we compare the $E_{abs550}$ before and during rain for 49 rain events from the yearlong measurements. Precipitation of 49 rain events as well as the monthly distributions are shown in Figure R7a. As for subtropical region, precipitation events are dominated in spring and summer time. EC concentration is only 43% during rain comparing before rain. However, as shown in Figure R7c, $E_{abs550}$ values are similar before and during rain as indicated by the unity slope. These results imply that $E_{abs550}$ is not very sensitive to rain effect in this study.

[Figure]

Figure R-7 Effect of precipitations. (a) Precipitations of 49 rain events and monthly distributions. (b) Scatter plot of EC before and during rain events. (c) Scatter plot of $E_{abs550}$ before and during rain events.

**R2-Q23.** 375-376: Could the authors clarify the reasoning given here? First, I believe the authors mean the $MAE_p$ determined in section 4.1, not 4.3, correct? I think this is arguing that because $MAE_p$ is, on average, higher than the MAE observed in the absence of BB influence there must be a large amount of BB influence on BC at this location? If so, I think some caution or caveats should be included in the discussion, since the BB influence tracer is not independent of the EC measurement. Further, I would expect there to be a much stronger effect on AAE if BB was such an important BC source, yet the later sections establish that AAE does not show a response.

**Author's Response:** Thanks for suggestion. The corresponding content has been revised as follows:

During the rainy season when oceanic wind prevails, BC from BB emission in Southeast Asia can reach PRD through long range transport (LRT), resulting in an elevated $K^+$/EC ratio and $MAE_{550}$. The Deming regression intercept (11.89) in Figure 8 represents the MAE without the BB effect. This non-BB $MAE_{550}$ (11.89 $m^2 g^{-1}$) is only slightly lower than $MAE_{p,550}$ (13 $m^2 g^{-1}$) obtained in section 4.3, implying that a large fraction of $MAE_{p,550}$ could not be explained by the BB source. Additional evidence was obtained through examining regression relationships of $MAE_{p,550}$ with $K^+$/EC month-by-month (Figure S17b). Correlation of monthly $MAE_{p,550}$ vs. $K^+$/EC ratio yield a $R^2$ of 0.23 (Figure S17c). In contrast, a much higher correlation ($R^2$=0.58) was observed between $MAE_{p,550}$ and non-BB $MAE_{550}$ (i.e., $K^+$/EC intercepts from Figure S17b). These results imply that BB is one of the contributors to the $MAE_{p,550}$ variations, but unlikely the dominating one.

**R2-Q24.** 381: Again I am curious as to the reasoning for selection of these wavelength pairs.

**Author's Response:** The 470 and 660 nm wavelength pairs were used to represent AAE in the visible range. $AAE_{370-880}$ (1.13±0.13) is added as shown in Figure S8b, which is slightly higher than $AAE_{470-660}$ (1.09±0.13) shown in FigureS8a.

**R2-Q25.** 384-386: I do not quite follow the reasoning presented here. Why does the monthly average of 1.2 suggest variations in AAE are associated with thicker coatings than BrC contribution? To me the strongest evidence of minimal BB contribution is the lack of correlation with the $K^+$/EC tracer. Does AAE show any correlation with $E_{abs}$? The manuscript asserts AAE is dominated by coating, so there should be a relationship based on the arguments presented.

**Author's Response:** We fully agree that the independency between AAE and $K^+$/EC is the main evidence that BB is not the driving force of AAE variation. The mention of monthly AAE of 1.2 here is another evidence to support this argument. The following content is revised to improve the clarity.

These results suggest that the elevated AAE observed in the PRD wintertime is unlikely to be dominated by the BB effect. Beside the independency between $AAE_{470-660}$ and $K^+$/EC ratio, the measured $AAE_{470-660}$ range also implies that BB is not the major driving force of $AAE_{470-660}$ variations. The limited light absorption contribution from BrC in RPD region is observed in a recent study (Yuan et al., 2016) , which suggest an upper limit of BrC contribution of 10% at 405 nm in the winter time using the AAE approach.

**R2-Q26.** 413-415: The refractive index of the shell also changes as it takes up water. Is this accounted for in the modeling?

**Author's Response:** The RI change due to water uptake is not considered in our Mie simulations.

**R2-Q27.** 426: This approach requires that only BC and associated coatings affect Eabs and AAE, but AAE can also be affected by non-BC aerosol (e.g., brown carbon or dust) that is externally mixed with BC. I am not sure how this approach can work unless there is clear evidence that there are no other light-absorbing particles or that the relative abundance does not change with season.

**Author's Response:** We add a section to discuss the caveats of the MRS method for $E_{abs}$ determination. The MRS approach is a tracer based method. For a scenario that samples are strongly influenced by BrC (e.g. sample overall AAE>2), it is possible to determine the contribution of $\sigma_{abs,Brc}$ if a reliable primary BrC tracer is available. We add the following content to discuss such scenarios.

The data in this study is dominated by BC absorption that did not show much influence from BrC. However, extra care should be taken if the samples exhibit substantial BrC signature (e.g. AAE>2). Such situations are equivalent to the two-source scenarios discussed in our previous paper on the MRS method (Wu and Yu, 2016) and the major findings are described below. Two types of two-source scenarios are considered: two correlated primary sources (scenario A) and two independent primary sources (scenario B). In scenario A in which both BC and primary BrC are dominated by BB, using BC as a solo tracer to calculate the primary ratio ($MAE_p$) still works. In scenario B in which BC and primary BrC are independent, using BC alone to determine a single primary $MAE_p$ could lead to a considerable bias in $E_{abs}$ estimation. Alternatively, if a reliable primary BrC tracer is available, the corresponding $MAE_{p,BrC}$ can be determined by MRS. With the knowledge of $MAE_{p,BrC}$ and $MAE_{p,BC}$, light absorption by BC and BrC can be calculated separately and the $E_{abs}$ can be determined using Eq. (11):

$$E_{abs} = \frac{\sigma_{abs,t}}{\sigma_{abs,p,BC}+\sigma_{abs,p,BrC}} = \frac{\sigma_{abs,t}}{MAE_{p,BC}\times EC+MAE_{p,BrC}\times BrC} \tag{11}$$

However, the implementation of Eq.11 is challenging due to the complexity in the chemical composition of BrC. For example, a recent study found that the 20 most absorbing BrC chromophores account for ~50% BrC light absorption and there is not a single compound contributing more than 10% (Lin et al., 2016), making it difficult to choose a single compound as the BrC tracer. In addition, time resolved measurement of BrC chromophores has yet to emerge. As a result, for scenario B (sample AAE>2 & primary BrC variations independent of BC), estimation of $E_{abs}$ by MRS is not practical at this stage due to the lack of required input data. Using BC alone to determine a single primary $MAE_p$ could lead to a considerable bias and should be avoided.

**R2-Q28.** Table 1, while convenient, could be omitted for length.

**Author's Response:** We feel that keeping table 1 in main text would be helpful for quick lookup of the abbreviations used in this study.

**R2-Q29.** Figures 2-4 do not add much to the manuscript, in my opinion, and could be removed for length.

**Author's Response:** Figure 2 is now moved to SI. As for Figure 3 and 4, they are useful to help the readers to understand how core/shell combinations can affect the variability of AAE and $E_{abs}$ form a theoretical perspective. These two figures are used in the later discussions in section 4.5 and 4.6. Keeping these two figures will be convenient for readers to understand how observed $E_{abs}$ and AAE can be used to infer the core\shell range from Figure 3 and 4. We believe the inclusion of the two Mie simulation figures is helpful in understand the linkage between observed AAE/$E_{abs}$ and core\shell ranges, especially from the cross-section plots.

**R2-Q30.** Figure 7: Minor, but x-axis is year and month, not just month. Replacing with "Date" would be fine.

**Author's Response:** Revision made.

[revised manuscript text omitted]
 | $\lambda$ (nm) | $\sigma_{abs}$ Instrument | EC determination protocol | $\sigma_{abs}$ ±1 S.D. (Mm⁻¹) | EC mass ($\mu$g m⁻³) | estimated $MAE_p$* (m² g⁻¹) | observed MAE (m² g⁻¹) arithmetic mean ±1 S.D. | Gaussian fit | Reference |
|---|---|---|---|---|---|---|---|---|---|---|---|---|
| Guangzhou, China | Suburban | 2012.2-2013.1 | PM2.5 | 550 | AE | NIOSH_TOT | 42.65±29.41 | 2.66±2.27 | 13* | 18.75±6.16 | 16.16 | This study |
| Shenzhen, China | Urban | 2011.8-9 | PM2.5 | 532 | PAS | LII | 25.4±19.0 | 4.0±3.1 | / | 6.5±0.5[6.29±0.48] | / | (Lan et al., 2013) |
| Xi'an, China | Urban | 2012.12-2013.1 | PM2.5 | 870 | PAS | LII | / | 8.8±7.3 | 7.17[11.34] | / | 7.62[12.05] | (Wang et al., 2014) |
| Xi'an, China | Urban | 2013.2 | PM2.5 | 532 | PAS | LII | | | | 14.6±5.6 | 12.7 | (Wang et al., 2017) |
| Guangzhou, China | Urban | 2004.10 | PM2.5 | 532 | PAS | NIOSH_TOT | 91±60 | 7.1 | 7.7[7.44] | / | / | (Andreae et al., 2008) |
| Fresno, USA | Urban | 2005.8-9 | PM2.5 | 532 | PAS | IMPROVE_A_TOR / NIOSH_TOT | 5.06 | 1.01 / 0.58 | / | 6.1±2.5[5.9±2.42] / 9.3±2.4[8.99±2.32] | / | (Chow et al., 2009) |
| T1,Mexico city, Mexico | Suburban | 2006.3 | PM2.5 | 870 | PAS | NIOSH_TOT | / | / | / | 9.2~9.7***[14.55~15.34] | / | (Doran et al., 2007) |
| Tokyo, Japan | Suburban | 2005.8 | PM2.5 | 565 | PSAP | IMPROVE_A_TOR | 30.43±20.41 | 2.9±2.13 | 11±1 | / | / | (Naoe et al., 2009) |
| Pasadena, USA | Urban | 2010.5-6 | PM2.5 | 532 | AM | NIOSH_TOT | 3.8±3.4 | 0.6~0.7 | 5.7[5.51] | / | / | (Thompson et al., 2012) |
| Toronto, Canada | Urban | 2006.12-2007.1 | PM2.5 | 760 | PAS | NIOSH_TOT | / | / | 6.9~9.1**[9.53~12.57] | 9.3~9.9[12.85~13.68] | / | (Knox et al., 2009) |
| Toronto, Canada | Suburban | | | | | | 3~6 | 0.10~0.14 | / | 30~43[42.6~61.06] | / | |
| Windsor, Canada | Urban | 2007.8 | PM2.5 | 781 | PAS | LII | 4.4±2.9 | 0.27±0.23 | / | 16±1[22.72±1.42] | / | (Chan et al., 2011) |
| Ottawa, Canada | Urban | | | | | | 26±17 | 1.7±0.9 | / | 15±3[21.3±4.26] | / | |
| Beijing, China | Rural | 2005.3 | / | 550 | AE | NIOSH_TOT | / | / | 9.5 | 11.3 | / | (Yang et al., 2009) |
| Montseny, Spin | Rural (Mediterranean) | 2009.11-2010.10 | PM10 | 637 | MAAP | NIOSH_TOT | 2.8±2.2 | 0.271±0.215 | / | 10.4[12.04] | / | (Pandolfi et al., 2011) |
| Jungfraujoch, Switzerland | Rural (high alpine) | 2007.2-3 | / | 637 | MAAP | LII | / | / | / | 10.2±3.2[11.81±3.71] | / | (Liu et al., 2010) |
| Lin'an, China | Rural | 1999.11 | PM2.5 | 550 | PSAP | NIOSH_TOT | 23±14 | 3.4±1.7 | / | 8.6±7.0 | / | (Xu et al., 2002) |
| Jeju Island, Korea | Coastal Rural, (East China Sea) | 2001.4 | PM10 | 550 | PSAP | NIOSH_TOT | / | / | / | 12.6±2.6 | / | (Chuang et al., 2003) |
| Maldives | Oceanic rural | 1999.2-3 | PM3 | 550 | PSAP | EGA | 62±34 | 2.5±1.4 | 6.6 | 8.1 | / | (Mayol-Bracero et al., 2002) |

*Determined by Minimum R Squared method; ** Median values;

[revised manuscript text omitted]

**Figure S**9. Comparison of $E_{abs}$ from original data and systematically biased data (Test A). It should be noted that the $E_{abs}$ shown here is ratio of averages, which is different form the annual average $E_{abs}$ calculated from average of ratios.

[Figure]

**Figure S10.** Comparison of $E_{abs}$ from data using NIOSH EC and data using IMPROVE EC (Test B). It should be noted that the $E_{abs}$ shown here is ratio of averages, which is different form the annual average $E_{abs}$ calculated from average of ratios.

[Figure]

| Wavelength (nm) | 370 | 470 | 520 | 532 | 550 | 590 | 660 | 880 | 950 |
|---|---|---|---|---|---|---|---|---|---|
| $E_{abs}$ mean | 1.55 | 1.51 | 1.50 | 1.50 | 1.50 | 1.49 | 1.49 | 1.48 | 1.49 |
| $E_{abs}$ S.D. | 0.48 | 0.47 | 0.47 | 0.47 | 0.48 | 0.47 | 0.48 | 0.49 | 0.49 |

**Figure S11.** Spectrum annual average $E_{abs}$ from 370 to 950 nm.

[Figure]

**Figure S1**2. Measured monthly variations of SSA$_{525}$.

[Figure]

**Figure S13.** Hourly back trajectories for the past 72 hours calculated using NOAA's HYSPLIT model from Feb 2012 to Jan 2013. The color coding represents different months.

[Figure]

**Figure S14.** Total spatial variance (TSV) as a function of number of clusters in back trajectories clustering analysis.

[Figure]

**Figure S15.** Frequency distributions of $E_{abs550}$ by different air mass clusters. P is calculated by Wilcoxon-Mann-Whitney tests.

[Figure]

**Figure S16.** (a) Monthly contribution of each cluster. (b) Monthly E$_{abs550}$ of each cluster.

[Figure]

**Figure S17.** (a) Monthly variations of $K^+$/EC ratio from 2012 Feb to 2013 Jan at NC site. (b) Monthly regressions between $MAE_{550}$ and $K^+$/EC with slope in red, intercept in green and $R^2$ in purple. (c) regressions between monthly $MAE_{p,550}$ and $K^+$/EC. (d) regression between monthly $MAE_{p,550}$ and intercepts from (b).

[Figure]

**Figure S18.** Correlations of AAE with $K^+$/EC ratio (biomass burning indicator). (a) AAE from 370 – 470 nm. (b) AAE from 470 – 660 nm.

[Figure]

**Figure S19.** Annual frequency distribution of LWC/non-EC PM$_{2.5}$ mass fraction.

[Figure]

[Figure]

**Figure S20.** Size range of soot particles constrained by $E_{abs}$, $SSA_{525}$ and $AAE_{470-660}$ from measurements. (a) Clear shell scenario; (b) Brown shell scenario

[Figure]

[Figure]

**Figure S21.** MRS program written in Igro Pro (WaveMetrics, Inc. Lake Oswego, OR, USA). Available from https://sites.google.com/site/wuchengust.

[Figure]

[Figure]

**Figure S22.** Mie program written in Igro Pro (WaveMetrics, Inc. Lake Oswego, OR, USA). Available from https://sites.google.com/site/wuchengust.

[Figure]

[Figure]

**Figure S23.** Aethalometer data processing program written in Igro Pro (WaveMetrics, Inc. Lake Oswego, OR, USA). Available from https://sites.google.com/site/wuchengust.

[Figure]

[Figure]

**Figure S24.** Histbox program written in Igro Pro (WaveMetrics, Inc. Lake Oswego, OR, USA). Available from https://sites.google.com/site/wuchengust.

[Figure]

**Figure S25.** Scatter plot program written in Igro Pro (WaveMetrics, Inc. Lake Oswego, OR, USA). Available from https://sites.google.com/site/wuchengust.

---

## Author Comment (AC2) · 15 Oct 2017

**Point-by-point response to review comments on manuscript acp-2017-582 "Quantifying black carbon light absorption enhancement by a novel statistical approach"**

**By Cheng Wu et al.**

We thank the two anonymous reviewers for their constructive comments to improve the manuscript. Our point-by-point responses to the review comments are listed below. Changes to the manuscript are marked in blue in the revised manuscript. The marked manuscript is submitted together with this response document.

**Anonymous Referee #1**

**R1-Q1.** This paper proposed a new approach to characterize the light absorption enhancement of BC and reported its application to the observed results based on an aethalometer and a thermo-optical carbon analyzer in Pearl River Delta area. This manuscript includes sufficient originality, and the topic seems to fit the journal. However, there are several concerns on the accuracy of the light absorption measurements, definition of $E_{abs}$, and discussions on the observed optical properties. I believe that the points below should be addressed before considering the publication in ACP.

**Author's Response:** Thanks for the very insightful and detailed comments. Please see below for point-by-point response to reviewers' comments.

**General comments**

**R1-Q2.** 1) Accuracy of the aethalometer measurements. Although the authors used the correction scheme reported by Weingartner et al. (2003) to correct the artifacts due to aerosol loading, filter matrix and scattering effect, it is not clear that the light absorption for "coated-BC" can be accurately measured (or corrected), especially under "high RH" conditions. For example, Arnott et al (2003) reported that a filter-based photometer has large potential artifacts above 80% RH. The authors need to give more detailed information on accuracy of aethalometer measurements.

**Author's Response:** We add the following content in section 2.1 to elaborate the factors that can affect the accuracy of Aethalometer in details.

Besides these artifacts, RH is also a source of $\sigma_{abs}$ measurement uncertainty. Elevated RH is not only a driving force of increased $\sigma_{abs}$ due to the hygroscopic growth of particles, but also a factor affecting ambient $\sigma_{abs}$ measurements. Previous studies found $\sigma_{abs}$ by PAS exhibit a systematic decrease when RH>70% (Arnott et al., 2003;Kozlov et al., 2011). Water evaporation was found as the major cause for the biased PAS $\sigma_{abs}$ measurements under high RH (Raspet et al., 2003;Lewis et al., 2009;Langridge et al., 2013). Filter-based measurements are also affected under high RH conditions. For example, Arnott et al. (2003) observed erratic responses by particle soot absorption photometer (PSAP) as RH changed. The main reason is traced to the hydrophilic cellulose membrane, which serves to reinforce the quartz filter used in PSAP. The fibers can swell and shrink as RH changes, causing unwanted light attenuation signal. The PTFE-coated glass-fiber tape has become available since 2012 for the recent model of Aethalometer to minimize the RH interference (Magee-Scientific, 2017). A study by Schmid et al. (2006) reported dependency of PSAP $\sigma_{abs}$ on RH, but found negligible effect of RH on Aethalometer performance. It is also worth noting that RH in the Aethalometer optical chamber may be lower than the ambient RH due to the slightly elevated temperature inside the instrument. The magnitude of RH difference was found similar between different instruments: 20% for the Aethalometer (Schmid et al., 2006) and 15% for the nephelometer (Guyon et al., 2004). The RH in the Aethalometer optical chamber was not measured in this study. We expected its level to be slightly lower than the ambient RH. Cappa et al. (2008) found $\sigma_{abs}$ measurements by PSAP and PAS maintained a high linearity even under high RH conditions (65-91%). Inter-comparison studies demonstrated that with proper corrections, Aethalometer $\sigma_{abs}$ measurements agree well with those by PAS (Ajtai et al., 2011). During the inter-comparison study of an Aethalometer (AE-16) and a PAS in Guangzhou (Wu et al., 2009), good correlation was found ($R^2$=0.96) as shown in Figure S1. These comparison results imply that the Aethalometer results are linearly correlated with PAS measurements and RH has a limited interference on Aethalometer measurements.

The Aethalometer measurements correlated well with the PAS results during a field campaign in Guangzhou as evidenced by the high $R^2$ (0.96) shown in Figure R-1. The ambient RH during the comparison period ranged from 20% to 95%. Since the PAS inlet was equipped with a dryer while Aethalometer was sampling without a dryer, the high $R^2$ proved that RH did not affect the correlation between Aethalometer and PAS. The results proved that Aethalometer can achieve comparable precision of PAS for hourly resolution data. We add tests in section 4.1 to evaluate the robustness of MRS against systematic biases. The results show that MRS is insensitive to systematic biases and details can be found in the response to R2-Q5.

[Figure]

**Figure R-1.** Comparison of collocated Aethalometer and PAS at Guangzhou (Oct 2004). Both PAS and Aethalometer (AE-16) were equipped with PM$_{2.5}$ inlets. RH of the sampled air was controlled to be <45% for PAS. Aethalometer sampling was conducted without RH control.

**R1-Q3.** 2) Definitions of the MAE$_p$ and E$_{abs}$ The MAE$_p$ obtained by the method in this study is MAE value for primary emission source. I think that the MAE$_p$, which was obtained in this study, includes the effects of "lensing effect" due to co-emitted OC. Therefore, the definition of E$_{abs}$ values should be different with the $E_{abs}$ in the previous studies, at least, using TD technique. Detailed explanation on this point should be added.

**Author's Response:** Thanks for raising this point, which we did not make it clear in our manuscript. Yes, we agree with the reviewer that the nature of $MAE_{p,TD}$ is different from $MAE_p$ at emission source. The following contents are added to clarify this point.

In introduction:

It's also worth noting that $MAE_p$ by the TD approach is different from the $MAE_p$ at the emission source. First, the morphology of thermally denuded BC particles (compact aggregates) is different from that of freshly emitted BC particles (chain-like aggregates). Second, most of the coatings is removed with the TD denuded soot particles, but freshly emitted soot particles usually come with a thin coating of OC formed from condensation of OC vapors as the temperature drops from engine to the ambient air. As a result, the $MAE_p$ by TD approach is expected to be lower than the $MAE_p$ of emission source. In this sense, the TD approach may not be a prefect "time machine" to reverse the aging process for $E_{abs}$ determination.

In section 4.1:

As mentioned in section 1, the definition of $MAE_p$ by the TD approach is different from the $MAE_p$ of emission source. The TD $MAE_p$ is expected to be slightly lower than the $MAE_p$ of emission source. Therefore, the corresponding $E_{abs}$ are slightly different and it should be cautioned when comparing MRS-derived $E_{abs}$ with $E_{abs}$ by the TD approach and Mie simulations.

**R1-Q4.** 3) The effect of biomass burning

3-1) Lines 371-374: "During the rainy season when oceanic prevailing wind dominates, BC from BB emission in Southeast Asia can reach PRD through long range transport (LRT), resulting in an elevated $K^+/EC$ ratio and $MAE_{550}$, which might be a combination of a thicker coating when freshly emitted from BB sources and enhanced coating during LRT."

=> If so, I think the $MAE_p$ values should be higher in summer (rainy season). Please consider about this point.

**Author's Response:** BB is one aerosol source affecting $MAE_{p,550}$ but not the dominating source. As shown in Figure 8 in the revised manuscript, the data points are scattered and the $R^2$ (0.33) is relatively low. The intercept in Figure 8 represents the part of MAE that are not explained by $K^+/EC$ ratio. To further clarify this point, we calculate $K^+/EC$ ratio and $MAE_{550}$ - $K^+/EC$ regression intercept of each month, then we compare the correlation of $MAE_{p,550}$ vs. $K^+/EC$ ratio and $MAE_{p,550}$ vs. $K^+/EC$ intercept. As shown in Figure R-2, $MAE_{p,550}$ has higher correlation with $K^+/EC$ intercept ($R^2=0.58$) than with $K^+/EC$ ratio ($R^2=0.23$). These results imply that BB is one of the contributors to the $MAE_{p,550}$ variations, but unlikely to be the dominating one. As a result, in some summer months, the $MAE_{p,550}$ is higher than the winter months but some months are not. Other BC sources and BB combustion conditions may affect the overall $MAE_{p,550}$. The relevant content is revised as follow:

During the rainy season when oceanic wind prevails, BC from BB emission in Southeast Asia can reach PRD through long range transport (LRT), resulting in an elevated $K^+/EC$ ratio and $MAE_{550}$. The Deming regression intercept (11.89) in Figure 8 represents the MAE without the BB effect. This non-BB $MAE_{550}$ (11.89 $m^2\ g^{-1}$) is only slightly lower than $MAE_{p,550}$ (13 $m^2\ g^{-1}$) obtained in section 4.3, implying that a large fraction of

$MAE_{p,550}$ could not be explained by the BB source. Additional evidence was obtained through examining regression relationships of $MAE_{p,550}$ with $K^+/EC$ month-by-month (Figure S17b). Correlation of monthly $MAE_{p,550}$ vs. $K^+/EC$ ratio yield a $R^2$ of 0.23 (Figure S17c). In contrast, a much higher correlation ($R^2=0.58$) was observed (Figure S17d) between $MAE_{p,550}$ and non-BB $MAE_{550}$ (i.e., $K^+/EC$ intercepts from Figure S17b). These results imply that BB is one of the contributors to the $MAE_{p,550}$ variations, but unlikely to be the dominating one.

[Figure]

**Figure R-2.** (a) Monthly variations of $K^+/EC$ ratio from 2012 Feb to 2013 Jan at NC site. (b) Monthly regressions between $MAE_{550}$ and $K^+/EC$ with slope in red, intercept in green and $R^2$ in purple. (c) regressions between monthly $MAE_{p,550}$ and $K^+/EC$. (d) regression between monthly $MAE_{p,550}$ and intercepts from (b).

**R1-Q5.** 3-2) Lines 384-388: "Since the monthly average AAE in wintertime didn't exceed 1.2 (Table S3), the variations of AAE in the PRD are likely more associated with thicker coatings rather than the contribution of BrC. The results also imply that attempts on BrC absorption attribution for the PRD dataset presented in this study could be risky, considering that elevation of AAE is actually dominated by coating (Lack and Langridge, 2013)."

=> Because higher $E_{abs}$ values were observe in summer, the coating would be thicker in summer. The suggestion that "the higher AAE in winter are likely more associated with thicker coatings" is not reasonable. Please consider about this point.

**Author's Response:** The point we intend to make here is that since the seasonal variations of $AAE_{470-660}$ is small, such variations are likely induced by the coating effect as shown in the Mie simulation rather than the presence of BrC. The corresponding text is rephrased as below to improve the clarity.

> Since the monthly average $AAE_{470-660}$ in wintertime did not exceed 1.2 (Table S3), the variations of $AAE_{470-660}$ in the PRD are more likely associated with coatings rather than the contribution of BrC.

**R1-Q6.** 4) The effect of relative humidity (RH) on optical properties As mentioned above, the authors need to show some evidences suggesting that the obtained positive correlation between f(RH) (i.e., MAE) and RH is not due to the artifact of RH on aethalometer measurements. If the measurements are assumed to be accurate, the observed higher $E_{abs}$ (and MAE) and lower AAE values in summer may be explained only by RH, considering that the RH may be higher in summer and RH for air masses from Southeast Asia. Which one of higher OC/EC ratio and higher RH do you suggest as a main contributor for observed higher $E_{abs}$ in summer?

**Author's Response:** The study by Schmid et al. (2006) found PSAP $\sigma_{abs}$ shows dependency on RH, but the effect of RH on Aethalometer performance is neglectable. See our response to R1-Q2 for more detailed discussion on RH effect.

The contribution of elevated OC/EC ratio to the increase of $E_{abs}$ might not be as important as RH in the summer time. First, the OC/EC ratios in summer is only slightly higher than other seasons (Figure R-3a). Second, as shown by the $f(OC/EC)_{MAE}$ plot (Figure R-3b), OC/EC induced MAE enchantment only occurred when OC/EC>4, which corresponded to the data with the highest 20% OC/EC ratio. As a result, the OC/EC induced MAE enhancement was only important for episodes of high OC/EC hours, which have a limited temporal coverage.

[Figure]

**Figure R-3.** (a) Monthly variations of OC/EC (b) $MAE_{550}$ enhancement as a function of OC/EC.

**Specific comments**

**R1-Q7.** 1) Line 235-236 "Thus, a BrC coating (brown shell) scenario is also considered in Mie simulation following the wavelength dependent RI suggested by Lack and Cappa (2010)." => Because the results of Mie simulation should be varied depending on the RI values given here, I recommend adding the range of RI values.

**Author's Response:** Suggestion taken. The corresponding content is revised as follows:

Thus, a BrC coating (brown shell) scenario is also considered in Mie simulation following the wavelength-dependent RI suggested by Lack and Cappa (2010), which ranges from 1.55-0.059i (370 nm) to 1.55-0.0005i (950 nm).

**R1-Q8.** 2) Section 3.2.4, 1st paragraph. For the brown shell scenario, I think that both of lensing effect and light absorption by BrC contribute to the $E_{abs}$ values. I recommend adding the fraction of each contribution.

**Author's Response:** We agree with the reviewer on this point. Fractional contribution from BrC is discussed as shown below. The following figure is added in SI (Figure S7).

[Figure]

**Figure R-4.** Mie simulated BrC absorption contribution to total $E_{abs}$ (lensing effect + BrC absorption) in the brown shell scenario. (a) 370 nm (b) 550 nm (c) 880 nm.

The following text is added in the revised manuscript (section 3.2.4):

The main reason behind is that in the brown shell scenario, both lensing effect and BrC absorption contribute to $E_{abs}$. As shown in Figure S7, the BrC absorption contribution to total $E_{abs}$ strongly depends on coating thickness and is insensitive to soot core diameters. When the coating is relatively thin (<5 nm for λ@370 nm, <15 nm for λ@550 nm and

<40 nm for λ@880 nm), BrC absorption contribution to the total $E_{abs}$ is less than 20%. As the coating increases to a certain level (~15 nm for λ@370 nm, ~35 nm for λ@550 nm and ~90 nm for λ@880 nm), BrC absorption contribution is comparable to the lensing effect contribution, each contributing ~50% to the total $E_{abs}$. When the BrC coating is sufficiently thick (>30 nm for λ@370 nm, >90 nm for λ@550 nm and >110 nm for λ@880 nm), BrC absorption dominates the $E_{abs}$ contribution. As a result, if BrC coating is indeed present in ambient samples, a strong wavelength dependent $E_{abs}$ could be observed, since a BrC coating of 30 nm would be enough to induce a large amount of detectable $E_{abs}$ in the UV range.

**R1-Q9.** 3) Section 4.1, 2nd paragraph The annual average $MAE_p$ value (13 $m^2/g$) is larger than those estimated for bare BC. It may indicate that the obtained MAEp values include the effects of "lensing effect" due to co-emitted OC, as mentioned above. If so, caution on the definition of MAEp in each study should be taken during comparison with other studies. Also, it is better to add information on the wavelength used in each study.

**Author's Response:** The authors fully agree with the reviewer that the difference in $MAE_p$ definition should be emphasized. Following contents are added in section 4.1 to clarify this point.

MAE$_p$ by MRS represents the MAE$_p$ at the emission source, which is different from the MAE$_p$ by the TD approach for two reasons. First, the morphology of thermally denuded BC particles (compact aggregates) is different from that of freshly emitted BC particles (chain-like aggregates). Second, most of the coatings are removed for TD denuded soot particles, but freshly emitted soot particles usually come with a thin coating of OC formed from condensation of OC vapors as the temperature drops from the flame to the ambient air. As a result, the MRS-derived MAE$_p$ is expected to be higher than the MAE$_p$ by the TD approach.

The MAE mentioned from literature was scaled to 550 nm for comparison. The following text is added to improve the clarity.

For comparison purpose, MAE measured at original wavelength and MAE scaled to 550 nm following the $λ^{-1}$ assumption are both shown in Table S1. The MAE comparisons discussed below are MAE at 550 nm.

**R1-Q10.** 4) Section 4.5 How did the absorption coefficients for dry conditions determine to obtained f(RH)?

**Author's Response:** The sampled air into the Aethalometer was not dried in this study. As the sampling covered a whole year, the RH also spanned a range sufficiently large to study the RH effect. The MAE data obtained at RH=30% (as shown in Figure 9a of the revised manuscript, the starting point of the f(RH) curve) are used as the dry condition to calculate f(RH).

**R1-Q11.** 5) Figure 9 What do to want to suggest from 30:1 line in Fig. 9?

[revised manuscript text omitted]

Fresh soot aggregates        Aged soot with coating        Simplified model in Mie

**Figure S2.** Schematic of the aging effect on light absorption. More light is absorbed by the soot particle core due to the lensing effect of the coating materials.

[Figure]

**Figure S3.** Mie simulated AAE$_{470-660}$ of a bare soot particle as a function of diameter with a Refractive index of 1.85 – 0.71i.

[Figure]

**Figure S4.** Mie simulated size dependency of soot particles SSA at wavelength 525 nm. (a)Combination of different clear shell (y axis) and core diameters (x axis). The color coding represents the SSA of a particle with specific core and clear shell size; (b) Cross-sections views of (a). The color coding represents different $D_{core}$ in the range of 50 – 300 nm. (c)&(d) Similar to (a)&(b) but from the brown shell scenario.

[Figure]

**Figure S5.** Mie simulated mass absorption efficiency (MAE$_p$) of a bare soot particle as a function of diameter at a wavelength of 550nm. Refractive index is 1.85 – 0.71i and density varied from 1.6 to 1.9 g cm$^{-3}$.

[Figure]

**Figure S6.** Mie simulated mass absorption efficiency (MAE) of a bare soot particle as a function of diameter at a wavelength of 550nm. Refractive index is 1.85 – 0.71i and density is 1.9 g cm$^{-3}$ for the soot core. Refractive index for clear coating is 1.55. Refractive index for brown coating is wavelength dependent adopted from Lack and Cappa (2010).

[Figure]

**Figure S7.** Mie simulated BrC absorption contribution to total $E_{abs}$ (lensing effect + BrC absorption) in the brown shell scenario. (a) 370 nm (b) 550 nm (c) 880 nm.

[Figure]

**Figure S8.** Measured annual statistics of AAE and SSA. (a) Annual frequency distribution of AAE$_{470-660}$. (b) Annual frequency distribution of AAE$_{370-880}$. (c) Annual frequency distribution of SSA$_{525}$. The red line represents lognormal fitting curve.

[Figure]

**Figure S**9. Comparison of $E_{abs}$ from original data and systematically biased data (Test A). It should be noted that the $E_{abs}$ shown here is ratio of averages, which is different form the annual average $E_{abs}$ calculated from average of ratios.

[Figure]

**Figure S10.** Comparison of $E_{abs}$ from data using NIOSH EC and data using IMPROVE EC (Test B). It should be noted that the $E_{abs}$ shown here is ratio of averages, which is different form the annual average $E_{abs}$ calculated from average of ratios.

[Figure]

| Wavelength (nm) | 370 | 470 | 520 | 532 | 550 | 590 | 660 | 880 | 950 |
|---|---|---|---|---|---|---|---|---|---|
| $E_{abs}$ mean | 1.55 | 1.51 | 1.50 | 1.50 | 1.50 | 1.49 | 1.49 | 1.48 | 1.49 |
| $E_{abs}$ S.D. | 0.48 | 0.47 | 0.47 | 0.47 | 0.48 | 0.47 | 0.48 | 0.49 | 0.49 |

**Figure S11.** Spectrum annual average $E_{abs}$ from 370 to 950 nm.

[Figure]

**Figure S1**2. Measured monthly variations of SSA$_{525}$.

[Figure]

**Figure S13.** Hourly back trajectories for the past 72 hours calculated using NOAA's HYSPLIT model from Feb 2012 to Jan 2013. The color coding represents different months.

[Figure]

**Figure S14.** Total spatial variance (TSV) as a function of number of clusters in back trajectories clustering analysis.

[Figure]

**Figure S15.** Frequency distributions of $E_{abs550}$ by different air mass clusters. P is calculated by Wilcoxon-Mann-Whitney tests.

[Figure]

**Figure S16.** (a) Monthly contribution of each cluster. (b) Monthly E$_{abs550}$ of each cluster.

[Figure]

**Figure S17.** (a) Monthly variations of $K^+$/EC ratio from 2012 Feb to 2013 Jan at NC site. (b) Monthly regressions between $MAE_{550}$ and $K^+$/EC with slope in red, intercept in green and $R^2$ in purple. (c) regressions between monthly $MAE_{p,550}$ and $K^+$/EC. (d) regression between monthly $MAE_{p,550}$ and intercepts from (b).

[Figure]

**Figure S18.** Correlations of AAE with $K^+$/EC ratio (biomass burning indicator). (a) AAE from 370 – 470 nm. (b) AAE from 470 – 660 nm.

[Figure]

**Figure S19.** Annual frequency distribution of LWC/non-EC PM$_{2.5}$ mass fraction.

[Figure]

[Figure]

**Figure S20.** Size range of soot particles constrained by $E_{abs}$, $SSA_{525}$ and $AAE_{470-660}$ from measurements. (a) Clear shell scenario; (b) Brown shell scenario

[Figure]

[Figure]

**Figure S21.** MRS program written in Igro Pro (WaveMetrics, Inc. Lake Oswego, OR, USA). Available from https://sites.google.com/site/wuchengust.

[Figure]

[Figure]

**Figure S22.** Mie program written in Igro Pro (WaveMetrics, Inc. Lake Oswego, OR, USA). Available from https://sites.google.com/site/wuchengust.

[Figure]

[Figure]

**Figure S23.** Aethalometer data processing program written in Igro Pro (WaveMetrics, Inc. Lake Oswego, OR, USA). Available from https://sites.google.com/site/wuchengust.

[Figure]

[Figure]

**Figure S24.** Histbox program written in Igro Pro (WaveMetrics, Inc. Lake Oswego, OR, USA). Available from https://sites.google.com/site/wuchengust.

[Figure]

**Figure S25.** Scatter plot program written in Igro Pro (WaveMetrics, Inc. Lake Oswego, OR, USA). Available from https://sites.google.com/site/wuchengust.

---

## Author Response (AR2)

**Point-by-point response to review comments on manuscript acp-2017-582 "Quantifying black carbon light absorption enhancement by a novel statistical approach"**

**By Cheng Wu et al.**

We thank the two anonymous reviewers for their comments to improve the manuscript. Our point-by-point responses to the review comments are listed below. Changes to the manuscript are marked in blue in the revised manuscript. The marked manuscript is submitted together with this response document.

**Anonymous Referee #2**

**R2-Q1.** The authors have made a number of improvements to the manuscript and adequately addressed my original comments with two important exceptions, detailed below. If these can be addressed to the satisfaction of the editor I recommend publication in ACP.

The response to comments regarding the influence of RH on the Aethalometer measurements is not adequate. The response states that a previously observed strong correlation between a non-dried Aethalometer and a dried photoacoustic instrument remained high even under elevated RH conditions. First, a strong correlation does not mean one of the instruments is not biased. For example, an instrument biased by a constant factor of two will still be strongly correlated with a reference measurement. A bigger point is that the data shown in the response show no increase in absorption measured by the Aethalometer relative to a dried reference measurement for RH values approaching 100%. This means that f(RH) is flat (1.0) for the Guangzhou data and there is no LWC influence on absorption. There could be real reasons for a difference in f(RH) during the two studies, but if this is the case, the Guangzhou data cannot be used to validate the response of the Aethalometer, since the conditions are different. Until the response of the Aethalometer has been validated against a reference instrument at high relative humidity conditions and at a location where there is an RH impact on absorption, Aethalometer data alone has too many uncertainties to draw the types of conclusions made in Section 4.5.

**Author's Response:** We understand the concerns regarding to the RH impact on Aethalometer measurements. RH effect is challenging to account for with $\sigma_{abs}$ measurement by in-situ techniques including PAS and extinction-minus-scattering approach. Since the f(RH) of aerosol light absorption is a much smaller quantity compared to that of scattering, the instrument precision is crucial for the determination of light absorption f(RH). Considering that we do not have data to validate the response of Aethalometer against a reference instrument at high RH conditions, we now have removed section 4.5 in the revised manuscript. We are looking forward to the future advancement of $\sigma_{abs}$ instrumentation to provide more precise measurements under high RH. f(RH) of light absorption is an important factor that needs to be investigated in the future, when the proper instrumentation become available.

The point we would like to make by showing the PAS-Aethalometer comparison dataset is that, the non-dried Aethalometer measurement correlated well with a dried PAS measurement. We do not intend to imply that one of the instrument is not biased.

While we have decided note to include in this manuscript the discussion on the dependency of optical properties on RH, we would like to address the below question raised by the reviewer: if f(RH) does exist in the Aethalometer measurement, will it affect the correlation between a non-dried Aethalometer and a dried PAS? We conducted a test using synthesized data to answer the question and the details are described below.

First, $\sigma_{abs}$ by a dried PAS is generated (Figure R1a) following a lognormal distribution using Mersenne Twister (MT) pseudorandom number generator. Then $\sigma_{abs}$ by a dried Aethalometer is generated (Figure R1b), which is highly correlated with $\sigma_{abs}$ of a dried PAS (Slope=1, $R^2$=0.99).

[Figure]

Figure R1. Distribution of synthesized $\sigma_{abs}$. (a) Dried PAS and (b) Dried Aethalometer

Then RH is generated independently following a uniform distribution in the range of 30~100% as shown in Figure R2a. The corresponding non-dried Aethalometer $\sigma_{abs}$ is derived as shown in Figure R2c following the f(RH) (distributions are shown in Figure R2b).

[Figure]

Figure R2. Distribution of synthesized data. (a) RH; (b) f(RH); and (c) non-dried Aethalometer $\sigma_{abs}$.

Scatter plots with linear regression lines are shown in Figure R3. From the simulation perspective, the inclusion of f(RH) only decrease the $R^2$ by a small amount (from 0.99 to 0.97). As a result, the high $R^2$ (0.96) alone shown in Figure S1 does not necessarily lead to the conclusion that f(RH) is flat, instead, the slope indicates whether f(RH) is >1 or flat. Since the Dried Aethalometer vs. Dried PAS is not tested with collocated non-dried Aethalometer vs. Dried PAS in our previous field measurements, the f(RH) effect requires further investigation in the future.

[Figure]

Figure R3. Scatter plots of synthesized data. (a) Dried Aethalometer vs. Dried PAS. (b) non-dried Aethalometer vs. Dried PAS.

**R2-Q2.** The addition of the tests to investigate impact of bias on the results is helpful, however my original comments were meant to question effects of time-varying biases on the data that depended on the amount of "coating" present or other physical processes not related to the real enhancement of light absorption. I apologize for not being more clear. The authors should generate a test data set with varying EC and prescribed "coating" contributions to measured absorption and EC, then introduce a bias to both data sets that depends on the amount and/or contribution of coating. Also useful would be tests or demonstrations of how other phenomena could affect the MSR method (e.g., a period of dust impact that affects absorption but not EC, or affects EC through mineral catalysis of EC, evaporation and/or adsorption of semi-volatile material from and/or onto filters in both the Aethalometer or OC/EC method). I believe this is merited given the novelty of the approach and analysis.

**Author's Response:** To further address the impact of measurement biases, we have re-organized section 5 (Caveats of the MRS method in its applications to ambient data) into four parts, covering the following aspects:
5.1 Impact of measurement biases
5.2 Impact of semi-volatile organic carbon
5.3 Impact of mineral dust
5.4 Impact of BrC

The discussions on systematic bias (Test A&B) are moved from section 4.1 to section 5.1 (Impact of measurement biases). Test C is introduced in section 5.1 to investigate the impact of sample-dependent bias as a function of $E_{abs}$ as shown below.

Study by Cheng et al. (2016) found two distinct types of biomass smoke behave differently on the biases of filter based $\sigma_{abs}$ measurement. The bias in the first type can be explained by a nearly constant correction factor, which is similar to the situation discussed in Test A. The bias in the second type shows an apparent OC/EC dependence. Test C is carried out to investigate this situation, i.e., examining the impact of sample-dependent bias as a function of $E_{abs}$. Unlike the proportional bias in Test A and B that is the same for all data points, the bias in Test C depends on the $E_{abs550}$ of individual samples, which are parametrized by Eqs. (11) and (12).

$$\sigma'_{abs550} = \sigma_{abs550} + \sigma_{abs550} \times (k \times E_{abs550} - k) \qquad (11)$$
$$EC' = EC - EC \times (k \times E_{abs550} - k) \qquad (12)$$

As shown in Eqs. (11) and (12), the positive bias of $\sigma_{abs550}$ and negative bias of EC are proportional to $E_{abs550}$. The magnitude of $E_{abs550}$-dependent bias is regulated by the factor k. Since $\sigma'_{abs550}$ and EC' are biased in different directions, resulting a further amplification in MAE biases, which could be considered as the extreme case. As shown in Figure S20, for k=10% (corresponding to a bias of 10% when $E_{abs}$=2), the bias of MRS-derived $E_{abs}$ is very small (1%). For k=20%, the MRS-derived $E_{abs}$ changes from 1.44 to 1.66, leading to a bias of 15%. These results imply that if the measurement bias follows the same form as demonstrated in Test C, the bias is not negligible but still acceptable. If the impact only affects $\sigma_{abs}$ or EC rather than impacting both, the bias is expected to be smaller than the estimation shown in Test C.

It should be noted that the parameterization scheme shown in Eqs. (11) and (12) is only for demonstration purpose from a conceptual perspective and it does not necessarily represent the real-world measurements. There is a lack of quantitative understanding of this impact. For example, Lee et al. (2007) used artificially fabricated EC samples with OC coatings to evaluate the impact of coating on OC/EC analysis.

Biases were observed, nevertheless, the results were linearly correlated with the true OC and EC values with a high $R^2$ (>0.9), implying that the biases in that specific study were dominated by systematic biases rather than the coating-dependent bias. Further studies are needed to better characterize and parameterize this impact if filter-based techniques are used for $\sigma_{abs}$ and EC determination in the MRS approach.

To address the impact of semi-volatile material, following contents are added as section 5.2 (Impact of semi-volatile organic carbon)

**5.2 Impact of semi-volatile organic carbon**

Light absorption contribution due to semi-volatile organic carbon (SVOC) from wood combustion was reported to be negligible in the visible range and around 10-20% at 360 nm (Chen and Bond, 2010). On the other hand, OCEC analysis can be affected by SVOC (Subramanian et al., 2004), either by positive artifacts through adsorption of SVOC onto quartz filters, or by negative artifacts through evaporation of SVOC due to the gas-particle re-equilibrium down stream of VOC denuder. Positive artifacts can be minimized by the installation of a VOC denuder which is widely adopted in RT-OCEC measurements (Bae et al., 2004; Bauer et al., 2009). A typical negative artifact of 10% is expected and can be corrected by backup filters (Subramanian et al., 2004). There was evidence to show that SVOC could affect OC/EC split in thermal/optical analysis (Cheng et al., 2009). However, the bias in EC caused by the OC/EC split drift due to SVOC is systematic, making it falls into the scenario discussed in Test B. As a result, the impact from SVOC on E_abs estimation by MRS is expected to be small.

Section 5.3 is added to provide the recommendations for samples strongly influenced by mineral dust.

**5.3 Impact of mineral dust**

The presence of mineral dust (MD) could affect both $\sigma_{abs}$ and EC determination. If MD is externally mixed with soot particles, the light absorption from MD could be miscounted as $\sigma_{abs}$ enhancement, leading to the overestimation of of E_abs. If the light absorption signal from MD is sufficiently strong (e.g. AAE>2), $\sigma_{abs}$ by MD and BC can be separated by the AAE approach suggested by Fialho et al. (2005). Additionally, the presence of substantial MD in samples has several impacts on the EC determination by thermal analysis. First, if the samples are not pre-treated with acid, the carbonated carbon could be misidentified as EC, resulting over-estimation of EC (Chow et al., 1993). The acid treatment is only available for off-line OC/EC analysis and not yet practical for the RT-OCEC analyzer. Second, metal oxides in MD can lead to premature EC oxidation in the helium stage of OC/EC analysis, leading to underestimation of EC (Wang et al., 2010; Bladt et al., 2012). The lack of a parameterization scheme for correcting the EC loss due to MD makes it improper to use the biased EC as a primary tracer. For these reasons, $E_{abs}$ estimation by MRS is not recommended for samples strongly influenced by MD.

Discussion on the impact of BrC (added in the last revision) is now moved to section 5.4.

**References**

[revised manuscript text omitted]